# TRAINING DIFFUSION MODELS WITH REINFORCEMENT LEARNING

Kevin Black[*1]     Michael Janner[*1]     Yilun Du[2]     Ilya Kostrikov[1]     Sergey Levine[1]

[1] University of California, Berkeley     [2] Massachusetts Institute of Technology

{kvablack, janner, kostrikov, sergey.levine}@berkeley.edu     yilundu@mit.edu

## ABSTRACT

Diffusion models are a class of flexible generative models trained with an approximation to the log-likelihood objective. However, most use cases of diffusion models are not concerned with likelihoods, but instead with downstream objectives such as human-perceived image quality or drug effectiveness. In this paper, we investigate reinforcement learning methods for directly optimizing diffusion models for such objectives. We describe how posing denoising as a multi-step decision-making problem enables a class of policy gradient algorithms, which we refer to as denoising diffusion policy optimization (DDPO), that are more effective than alternative reward-weighted likelihood approaches. Empirically, DDPO can adapt text-to-image diffusion models to objectives that are difficult to express via prompting, such as image compressibility, and those derived from human feedback, such as aesthetic quality. Finally, we show that DDPO can improve prompt-image alignment using feedback from a vision-language model without the need for additional data collection or human annotation. The project's website can be found at http://rl-diffusion.github.io.

## 1 INTRODUCTION

Diffusion probabilistic models (Sohl-Dickstein et al., 2015) have recently emerged as the de facto standard for generative modeling in continuous domains. Their flexibility in representing complex, high-dimensional distributions has led to the adoption of diffusion models in applications including image and video synthesis (Ramesh et al., 2021; Saharia et al., 2022; Ho et al., 2022), drug and material design (Xu et al., 2021; Xie et al., 2021; Schneuing et al., 2022), and continuous control (Janner et al., 2022; Wang et al., 2022; Hansen-Estruch et al., 2023). The key idea behind diffusion models is to iteratively transform a simple prior distribution into a target distribution by applying a sequential denoising process. This procedure is conventionally motivated as a maximum likelihood estimation problem, with the objective derived as a variational lower bound on the log-likelihood of the training data.

However, most use cases of diffusion models are not directly concerned with likelihoods, but instead with downstream objective such as human-perceived image quality or drug effectiveness. In this paper, we consider the problem of training diffusion models to satisfy such objectives directly, as opposed to matching a data distribution. This problem is challenging because exact likelihood computation with diffusion models is intractable, making it difficult to apply many conventional reinforcement learning (RL) algorithms. We instead propose to frame denoising as a multi-step decision-making task, using the exact likelihoods at each denoising step in place of the approximate likelihoods induced by a full denoising process. We present a policy gradient algorithm, which we refer to as denoising diffusion policy optimization (DDPO), that can optimize a diffusion model for downstream tasks using only a black-box reward function.

We apply our algorithm to the finetuning of large text-to-image diffusion models. Our initial evaluation focuses on tasks that are difficult to specify via prompting, such as image compressibility, and those derived from human feedback, such as aesthetic quality. However, because many reward functions of interest are difficult to specify programmatically, finetuning procedures often rely on large-scale human labeling efforts to obtain a reward signal (Ouyang et al., 2022). In the case of text-to-image diffusion, we propose a method for replacing such labeling with feedback from a vision-language model (VLM). Similar to RLAIF finetuning for language models (Bai et al., 2022b), the resulting procedure allows for diffusion models to be adapted to reward functions that would otherwise require

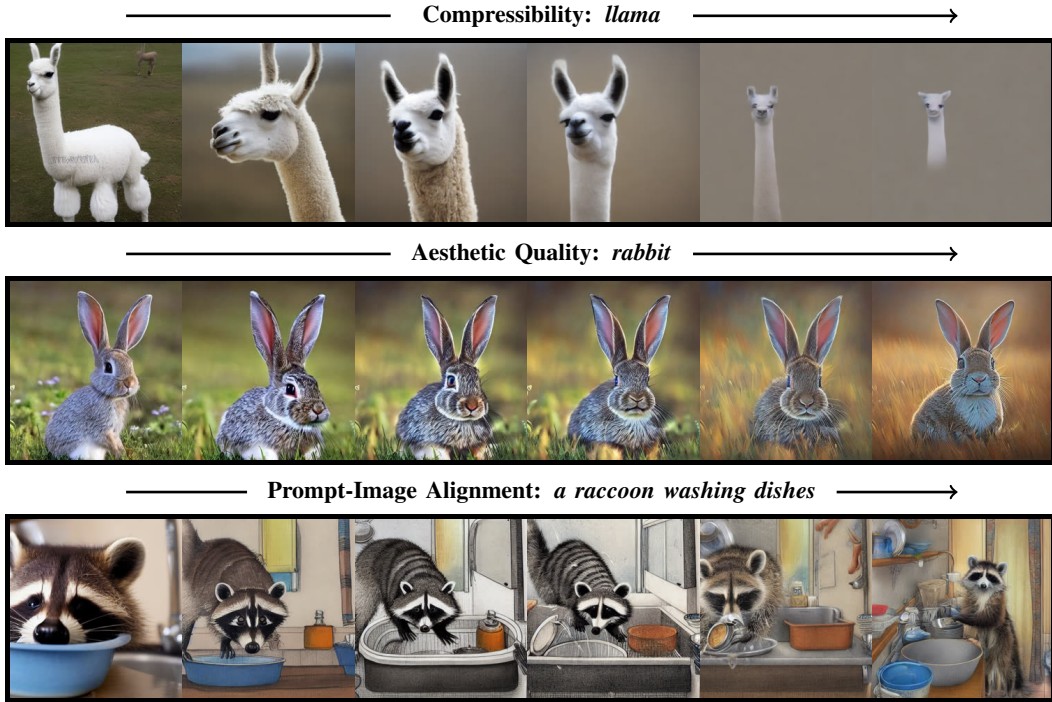

**Figure 1 (Reinforcement learning for diffusion models)** We propose a reinforcement learning algorithm, DDPO, for optimizing diffusion models on downstream objectives such as compressibility, aesthetic quality, and prompt-image alignment as determined by vision-language models. Each row shows a progression of samples for the same prompt and random seed over the course of training.

additional human annotations. We use this procedure to improve prompt-image alignment for unusual subject-setting compositions.

Our contributions are as follows. We first present the derivation and conceptual motivation of DDPO. We then document the design of various reward functions for text-to-image generation, ranging from simple computations to workflows involving large VLMs, and demonstrate the effectiveness of DDPO compared to alternative reward-weighted likelihood methods in these settings. Finally, we demonstrate the generalization ability of our finetuning procedure to unseen prompts.

## 2 RELATED WORK

**Diffusion probabilistic models.** Denoising diffusion models (Sohl-Dickstein et al., 2015; Ho et al., 2020) have emerged as an effective class of generative models for modalities including images (Ramesh et al., 2021; Saharia et al., 2022), videos (Ho et al., 2022; Singer et al., 2022), 3D shapes (Zhou et al., 2021; Zeng et al., 2022), and robotic trajectories (Janner et al., 2022; Ajay et al., 2022; Chi et al., 2023). While the denoising objective is conventionally derived as an approximation to likelihood, the training of diffusion models typically departs from maximum likelihood in several ways (Ho et al., 2020). Modifying the objective to more strictly optimize likelihood (Nichol & Dhariwal, 2021; Kingma et al., 2021) often leads to worsened image quality, as likelihood is not a faithful proxy for visual quality. In this paper, we show how diffusion models can be optimized directly for downstream objectives.

**Controllable generation with diffusion models.** Recent progress in text-to-image diffusion models (Ramesh et al., 2021; Saharia et al., 2022) has enabled fine-grained high-resolution image synthesis. To further improve the controllability and quality of diffusion models, recent approaches have investigated finetuning on limited user-provided data (Ruiz et al., 2022), optimizing text embeddings for new concepts (Gal et al., 2022), composing models (Du et al., 2023; Liu et al., 2022), adapters for additional input constraints (Zhang & Agrawala, 2023), and inference-time techniques such as classifier (Dhariwal & Nichol, 2021) and classifier-free (Ho & Salimans, 2021) guidance.

**Reinforcement learning from human feedback.** A number of works have studied using human feedback to optimize models in settings such as simulated robotic control (Christiano et al., 2017), game-playing (Knox & Stone, 2008), machine translation (Nguyen et al., 2017), citation retrieval (Menick et al., 2022), browsing-based question-answering (Nakano et al., 2021), summarization (Stiennon et al., 2020; Ziegler et al., 2019), instruction-following (Ouyang et al., 2022), and alignment with specifications (Bai et al., 2022a). Recently, Lee et al. (2023) studied the alignment of text-to-image diffusion models to human preferences using a method based on reward-weighted likelihood maximization. In our comparisons, their method corresponds to one iteration of the reward-weighted regresion (RWR) method. Our results demonstrate that DDPO significantly outperforms even multiple iterations of weighted likelihood maximization (RWR-style) optimization.

**Diffusion models as sequential decision-making processes.** Although predating diffusion models, Bachman & Precup (2015) similarly posed data generation as a sequential decision-making problem and used the resulting framework to apply reinforcement learning methods to image generation. More recently, Fan & Lee (2023) introduced a policy gradient method for training diffusion models. However, this paper aimed to improve data distribution matching rather than optimizing downstream objectives, and therefore the only reward function considered was a GAN-like discriminator. In concurrent work to ours, DPOK (Fan et al., 2023) built upon Fan & Lee (2023) and Lee et al. (2023) to better align text-to-image diffusion models to human preferences using a policy gradient algorithm. Like Lee et al. (2023), DPOK only considers a single preference-based reward function (Xu et al., 2023); additionally, their work studies KL-regularization and primarily focuses on training a different diffusion model for each prompt. In contrast, we train on many prompts at once (up to 398) and demonstrate generalization to many more prompts outside of the training set. Furthermore, we study how DDPO can be applied to multiple reward functions beyond those based on human feedback, including how rewards derived automatically from VLMs can improve prompt-image alignment. We provide a direct comparison to DPOK in Appendix C.

## 3   PRELIMINARIES

In this section, we provide a brief background on diffusion models and the RL problem formulation.

### 3.1   DIFFUSION MODELS

In this work, we consider conditional diffusion probabilistic models (Sohl-Dickstein et al., 2015; Ho et al., 2020), which represent a distribution $p(\mathbf{x}_0|\mathbf{c})$ over a dataset of samples $\mathbf{x}_0$ and corresponding contexts $\mathbf{c}$. The distribution is modeled as the reverse of a Markovian forward process $q(\mathbf{x}_t \mid \mathbf{x}_{t-1})$, which iteratively adds noise to the data. Reversing the forward process can be accomplished by training a neural network $\boldsymbol{\mu}_\theta(\mathbf{x}_t, \mathbf{c}, t)$ with the following objective:

$$\mathcal{L}_{\text{DDPM}}(\theta) = \mathbb{E}_{(\mathbf{x}_0, \mathbf{c}) \sim p(\mathbf{x}_0, \mathbf{c}), \, t \sim \mathcal{U}\{0, T\}, \, \mathbf{x}_t \sim q(\mathbf{x}_t | \mathbf{x}_0)} \left[ \| \tilde{\boldsymbol{\mu}}(\mathbf{x}_0, t) - \boldsymbol{\mu}_\theta(\mathbf{x}_t, \mathbf{c}, t) \|^2 \right] \quad (1)$$

where $\tilde{\boldsymbol{\mu}}$ is the posterior mean of the forward process, a weighted average of $\mathbf{x}_0$ and $\mathbf{x}_t$. This objective is justified as maximizing a variational lower bound on the log-likelihood of the data (Ho et al., 2020).

Sampling from a diffusion model begins with drawing a random $\mathbf{x}_T \sim \mathcal{N}(\mathbf{0}, \mathbf{I})$ and following the reverse process $p_\theta(\mathbf{x}_{t-1} \mid \mathbf{x}_t, \mathbf{c})$ to produce a trajectory $\{\mathbf{x}_T, \mathbf{x}_{T-1}, \ldots, \mathbf{x}_0\}$ ending with a sample $\mathbf{x}_0$. The sampling process depends not only on the predictor $\boldsymbol{\mu}_\theta$ but also the choice of sampler. Most popular samplers (Ho et al., 2020; Song et al., 2021) use an isotropic Gaussian reverse process with a fixed timestep-dependent variance:

$$p_\theta(\mathbf{x}_{t-1} \mid \mathbf{x}_t, \mathbf{c}) = \mathcal{N}(\mathbf{x}_{t-1} \mid \boldsymbol{\mu}_\theta(\mathbf{x}_t, \mathbf{c}, t), \sigma_t^2 \mathbf{I}). \quad (2)$$

### 3.2   MARKOV DECISION PROCESSES AND REINFORCEMENT LEARNING

A Markov decision process (MDP) is a formalization of sequential decision-making problems. An MDP is defined by a tuple $(\mathcal{S}, \mathcal{A}, \rho_0, P, R)$, in which $\mathcal{S}$ is the state space, $\mathcal{A}$ is the action space, $\rho_0$ is the distribution of initial states, $P$ is the transition kernel, and $R$ is the reward function. At each timestep $t$, the agent observes a state $\mathbf{s}_t \in \mathcal{S}$, takes an action $\mathbf{a}_t \in \mathcal{A}$, receives a reward $R(\mathbf{s}_t, \mathbf{a}_t)$, and transitions to a new state $\mathbf{s}_{t+1} \sim P(\mathbf{s}_{t+1} \mid \mathbf{s}_t, \mathbf{a}_t)$. An agent acts according to a policy $\pi(\mathbf{a} \mid \mathbf{s})$.

As the agent acts in the MDP, it produces trajectories, which are sequences of states and actions $\tau = (\mathbf{s}_0, \mathbf{a}_0, \mathbf{s}_1, \mathbf{a}_1, \ldots, \mathbf{s}_T, \mathbf{a}_T)$. The reinforcement learning (RL) objective is for the agent to maximize $\mathcal{J}_{\text{RL}}(\pi)$, the expected cumulative reward over trajectories sampled from its policy:

$$\mathcal{J}_{\text{RL}}(\pi) = \mathbb{E}_{\tau \sim p(\tau | \pi)} \left[ \sum_{t=0}^{T} R(\mathbf{s}_t, \mathbf{a}_t) \right].$$

## 4 REINFORCEMENT LEARNING TRAINING OF DIFFUSION MODELS

We now describe how RL algorithms can be used to train diffusion models. We present two classes of methods and show that each corresponds to a different mapping of the denoising process to the MDP framework.

### 4.1 PROBLEM STATEMENT

We assume a pre-existing diffusion model, which may be pretrained or randomly initialized. Assuming a fixed sampler, the diffusion model induces a sample distribution $p_\theta(\mathbf{x}_0 \mid \mathbf{c})$. The denoising diffusion RL objective is to maximize a reward signal $r$ defined on the samples and contexts:

$$\mathcal{J}_{\text{DDRL}}(\theta) = \mathbb{E}_{\mathbf{c} \sim p(\mathbf{c}), \, \mathbf{x}_0 \sim p_\theta(\mathbf{x}_0 | \mathbf{c})} [r(\mathbf{x}_0, \mathbf{c})]$$

for some context distribution $p(\mathbf{c})$ of our choosing.

### 4.2 REWARD-WEIGHTED REGRESSION

To optimize $\mathcal{J}_{\text{DDRL}}$ with minimal changes to standard diffusion model training, we can use the denoising loss $\mathcal{L}_{\text{DDPM}}$ (Equation 1), but with training data $\mathbf{x}_0 \sim p_\theta(\mathbf{x}_0 \mid \mathbf{c})$ and an added weighting that depends on the reward $r(\mathbf{x}_0, \mathbf{c})$. Lee et al. (2023) describe a single-round version of this procedure for diffusion models, but in general this approach can be performed for multiple rounds of alternating sampling and training, leading to an online RL method. We refer to this general class of algorithms as reward-weighted regression (RWR) (Peters & Schaal, 2007).

A standard weighting scheme uses exponentiated rewards to ensure nonnegativity,

$$w_{\text{RWR}}(\mathbf{x}_0, \mathbf{c}) = \frac{1}{Z} \exp \big( \beta r(\mathbf{x}_0, \mathbf{c}) \big),$$

where $\beta$ is an inverse temperature and $Z$ is a normalization constant. We also consider a simplified weighting scheme that uses binary weights,

$$w_{\text{sparse}}(\mathbf{x}_0, \mathbf{c}) = \mathbb{1}\big[ r(\mathbf{x}_0, \mathbf{c}) \geq C \big],$$

where $C$ is a reward threshold determining which samples are used for training. In supervised learning terms, this is equivalent to repeated filtered finetuning on training data coming from the model.

Within the RL formalism, the RWR procedure corresponds to the following one-step MDP:

$$\mathbf{s} \triangleq \mathbf{c} \qquad \mathbf{a} \triangleq \mathbf{x}_0 \qquad \pi(\mathbf{a} \mid \mathbf{s}) \triangleq p_\theta(\mathbf{x}_0 \mid \mathbf{c}) \qquad \rho_0(\mathbf{s}) \triangleq p(\mathbf{c}) \qquad R(\mathbf{s}, \mathbf{a}) \triangleq r(\mathbf{x}_0, \mathbf{c})$$

with a transition kernel $P$ that immediately leads to an absorbing termination state. Therefore, maximizing $\mathcal{J}_{\text{DDRL}}(\theta)$ is equivalent to maximizing the RL objective $\mathcal{J}_{\text{RL}}(\pi)$ in this MDP.

From RL literature, weighting a log-likelihood objective by $w_{\text{RWR}}$ is known to approximately maximize $\mathcal{J}_{\text{RL}}(\pi)$ subject to a KL divergence constraint on $\pi$ (Nair et al., 2020). However, $\mathcal{L}_{\text{DDPM}}$ (Equation 1) does not involve an exact log-likelihood — it is instead derived as a variational bound on $\log p_\theta(\mathbf{x}_0 \mid \mathbf{c})$. Therefore, the RWR procedure applied to diffusion model training is not theoretically justified and only optimizes $\mathcal{J}_{\text{DDRL}}$ very approximately.

### 4.3 DENOISING DIFFUSION POLICY OPTIMIZATION

RWR relies on an approximate log-likelihood because it ignores the sequential nature of the denoising process, only using the final samples $\mathbf{x}_0$. In this section, we show how the denoising process can be reframed as a *multi-step* MDP, allowing us to directly optimize $\mathcal{J}_{\text{DDRL}}$ using policy gradient

estimators. This follows the derivation in Fan & Lee (2023), who prove an equivalence between their method and a policy gradient algorithm where the reward is a GAN-like discriminator. We present a general framework with an arbitrary reward function, motivated by our desire to optimize arbitrary downstream objectives (Section 5). We refer to this class of algorithms as denoising diffusion policy optimization (DDPO) and present two variants based on specific gradient estimators.

**Denoising as a multi-step MDP.** We map the iterative denoising procedure to the following MDP:

$$\mathbf{s}_t \triangleq (\mathbf{c}, t, \mathbf{x}_t) \quad \pi(\mathbf{a}_t \mid \mathbf{s}_t) \triangleq p_\theta(\mathbf{x}_{t-1} \mid \mathbf{x}_t, \mathbf{c}) \qquad P(\mathbf{s}_{t+1} \mid \mathbf{s}_t, \mathbf{a}_t) \triangleq (\delta_{\mathbf{c}}, \delta_{t-1}, \delta_{\mathbf{x}_{t-1}})$$

$$\mathbf{a}_t \triangleq \mathbf{x}_{t-1} \qquad \rho_0(\mathbf{s}_0) \triangleq (p(\mathbf{c}), \delta_T, \mathcal{N}(\mathbf{0}, \mathbf{I})) \qquad R(\mathbf{s}_t, \mathbf{a}_t) \triangleq \begin{cases} r(\mathbf{x}_0, \mathbf{c}) & \text{if } t = 0 \\ 0 & \text{otherwise} \end{cases}$$

in which $\delta_y$ is the Dirac delta distribution with nonzero density only at $y$. Trajectories consist of $T$ timesteps, after which $P$ leads to a termination state. The cumulative reward of each trajectory is equal to $r(\mathbf{x}_0, \mathbf{c})$, so maximizing $\mathcal{J}_{\text{DDRL}}(\theta)$ is equivalent to maximizing $\mathcal{J}_{\text{RL}}(\pi)$ in this MDP.

The benefit of this formulation is that if we use a standard sampler with $p_\theta(\mathbf{x}_{t-1} \mid \mathbf{x}_t, \mathbf{c})$ parameterized as in Equation 2, the policy $\pi$ becomes an isotropic Gaussian as opposed to the arbitrarily complicated distribution $p_\theta(\mathbf{x}_0 \mid \mathbf{c})$ as it is in the RWR formulation. This simplification allows for the evaluation of exact log-likelihoods and their gradients with respect to the diffusion model parameters.

**Policy gradient estimation.** With access to likelihoods and likelihood gradients, we can make direct Monte Carlo estimates of $\nabla_\theta \mathcal{J}_{\text{DDRL}}$. Like RWR, DDPO alternates collecting denoising trajectories $\{\mathbf{x}_T, \mathbf{x}_{T-1}, \ldots, \mathbf{x}_0\}$ via sampling and updating parameters via gradient descent.

The first variant of DDPO, which we call DDPO$_{\text{SF}}$, uses the score function policy gradient estimator, also known as the likelihood ratio method or REINFORCE (Williams, 1992; Mohamed et al., 2020):

$$\nabla_\theta \mathcal{J}_{\text{DDRL}} = \mathbb{E}\left[ \sum_{t=0}^{T} \nabla_\theta \log p_\theta(\mathbf{x}_{t-1} \mid \mathbf{x}_t, \mathbf{c}) \, r(\mathbf{x}_0, \mathbf{c}) \right] \qquad (\text{DDPO}_{\text{SF}})$$

where the expectation is taken over denoising trajectories generated by the current parameters $\theta$.

However, this estimator only allows for one step of optimization per round of data collection, as the gradient must be computed using data generated by the current parameters. To perform multiple steps of optimization, we may use an importance sampling estimator (Kakade & Langford, 2002):

$$\nabla_\theta \mathcal{J}_{\text{DDRL}} = \mathbb{E}\left[ \sum_{t=0}^{T} \frac{p_\theta(\mathbf{x}_{t-1} \mid \mathbf{x}_t, \mathbf{c})}{p_{\theta_{\text{old}}}(\mathbf{x}_{t-1} \mid \mathbf{x}_t, \mathbf{c})} \, \nabla_\theta \log p_\theta(\mathbf{x}_{t-1} \mid \mathbf{x}_t, \mathbf{c}) \, r(\mathbf{x}_0, \mathbf{c}) \right] \qquad (\text{DDPO}_{\text{IS}})$$

where the expectation is taken over denoising trajectories generated by the parameters $\theta_{\text{old}}$. This estimator becomes inaccurate if $p_\theta$ deviates too far from $p_{\theta_{\text{old}}}$, which can be addressed using trust regions (Schulman et al., 2015) to constrain the size of the update. In practice, we implement the trust region via clipping, as in proximal policy optimization (Schulman et al., 2017).

## 5 REWARD FUNCTIONS FOR TEXT-TO-IMAGE DIFFUSION

In this work, we evaluate our methods on text-to-image diffusion. Text-to-image diffusion serves as a valuable test environment for reinforcement learning due to the availability of large pretrained models and the versatility of using diverse and visually interesting reward functions. In this section, we outline our selection of reward functions. We study a spectrum of reward functions of varying complexity, ranging from those that are straightforward to specify and evaluate to those that capture the depth of real-world downstream tasks.

### 5.1 COMPRESSIBILITY AND INCOMPRESSIBILITY

The capabilities of text-to-image diffusion models are limited by the co-occurrences of text and images in their training distribution. For instance, images are rarely captioned with their file size, making it impossible to specify a desired file size via prompting. This limitation makes reward functions based on file size a convenient case study: they are simple to compute, but not controllable through the conventional methods of likelihood maximization and prompt engineering.

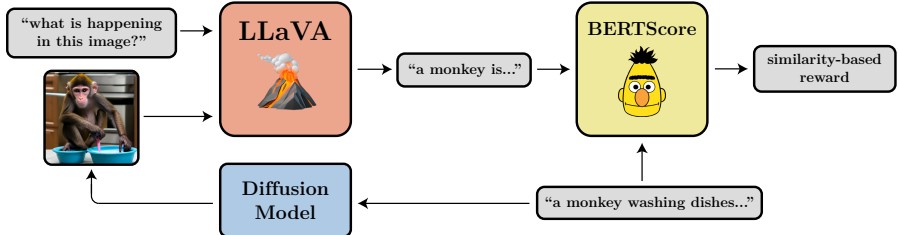

**Figure 2 (VLM reward function)** Illustration of the VLM-based reward function for prompt-image alignment. LLaVA (Liu et al., 2023) provides a short description of a generated image; the reward is the similarity between this description and the original prompt as measured by BERTScore (Zhang et al., 2020).

We fix the resolution of diffusion model samples at 512x512, such that the file size is determined solely by the compressibility of the image. We define two tasks based on file size: compressibility, in which the file size of the image after JPEG compression is minimized, and incompressibility, in which the same measure is maximized.

## 5.2   AESTHETIC QUALITY

To capture a reward function that would be useful to a human user, we define a task based on perceived aesthetic quality. We use the LAION aesthetics predictor (Schuhmann, 2022), which is trained on 176,000 human image ratings. The predictor is implemented as a linear model on top of CLIP embeddings (Radford et al., 2021). Annotations range between 1 and 10, with the highest-rated images mostly containing artwork. Since the aesthetic quality predictor is trained on human judgments, this task constitutes reinforcement learning from human feedback (Ouyang et al., 2022; Christiano et al., 2017; Ziegler et al., 2019).

## 5.3   AUTOMATED PROMPT ALIGNMENT WITH VISION-LANGUAGE MODELS

A very general-purpose reward function for training a text-to-image model is prompt-image alignment. However, specifying a reward that captures generic prompt alignment is difficult, conventionally requiring large-scale human labeling efforts. We propose using an existing VLM to replace additional human annotation. This design is inspired by recent work on RLAIF (Bai et al., 2022b), in which language models are improved using feedback from themselves.

We use LLaVA (Liu et al., 2023), a state-of-the-art VLM, to describe an image. The finetuning reward is the BERTScore (Zhang et al., 2020) recall metric, a measure of semantic similarity, using the prompt as the reference sentence and the VLM description as the candidate sentence. Samples that more faithfully include all of the details of the prompt receive higher rewards, to the extent that those visual details are legible to the VLM.

In Figure 2, we show one simple question: "*what is happening in this image?*". While this captures the general task of prompt-image alignment, in principle any question could be used to specify complex or hard-to-define reward functions for a particular use case. One could even employ a language model to automatically generate candidate questions and evaluate responses based on the prompt. This framework provides a flexible interface where the complexity of the reward function is only limited by the capabilities of the vision and language models involved.

## 6   EXPERIMENTAL EVALUATION

The purpose of our experiments is to evaluate the effectiveness of RL algorithms for finetuning diffusion models to align with a variety of user-specified objectives. After examining the viability of the general approach, we focus on the following questions:

1. How do variants of DDPO compare to RWR and to each other?
2. Can VLMs allow for optimizing rewards that are difficult to specify manually?
3. Do the effects of RL finetuning generalize to prompts not seen during finetuning?

**Pretrained**

**Aesthetic Quality**

**Compressibility**

**Incompressibility**

**Figure 3 (DDPO samples)** Qualitative depiction of the effects of RL finetuning on different reward functions. DDPO transforms naturalistic images into stylized artwork to maximize aesthetic quality, removes background content and applies foreground smoothing to maximize compressibility, and adds high-frequency noise to maximize incompressibility.

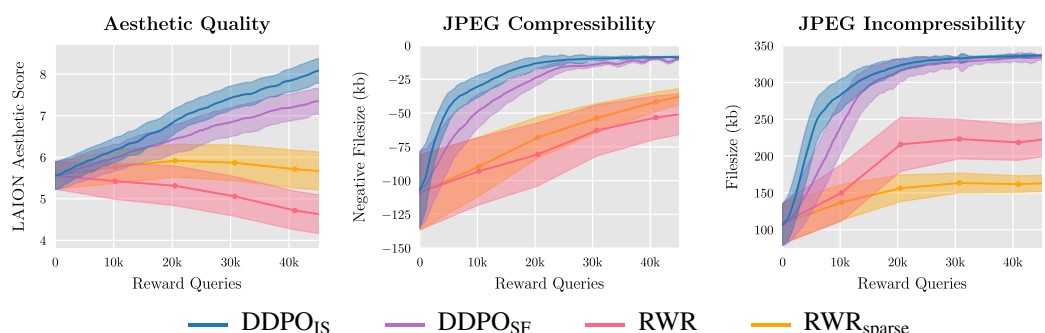

**Figure 4 (Finetuning effectiveness)** The relative effectiveness of different RL algorithms on three reward functions. We find that the policy gradient variants, denoted DDPO, are more effective optimizers than both RWR variants.

## 6.1 ALGORITHM COMPARISONS

We begin by evaluating all methods on the compressibility, incompressibility, and aesthetic quality tasks, as these tasks isolate the effectiveness of the RL approach from considerations relating to the VLM reward function. We use Stable Diffusion v1.4 (Rombach et al., 2022) as the base model for all experiments. Compressibility and incompressibility prompts are sampled uniformly from all 398 animals in the ImageNet-1000 (Deng et al., 2009) categories. Aesthetic quality prompts are sampled uniformly from a smaller set of 45 common animals.

As shown qualitatively in Figure 3, DDPO is able to effectively adapt a pretrained model with only the specification of a reward function and without any further data curation. The strategies found to optimize each reward are nontrivial; for example, to maximize LAION-predicted aesthetic quality, DDPO transforms a model that produces naturalistic images into one that produces artistic drawings. To maximize compressibility, DDPO removes backgrounds and applies smoothing to what remains. To maximize incompressibility, DDPO finds artifacts that are difficult for the JPEG compression algorithm to encode, such as high-frequency noise and sharp edges. Samples from RWR are provided in Appendix G for comparison.

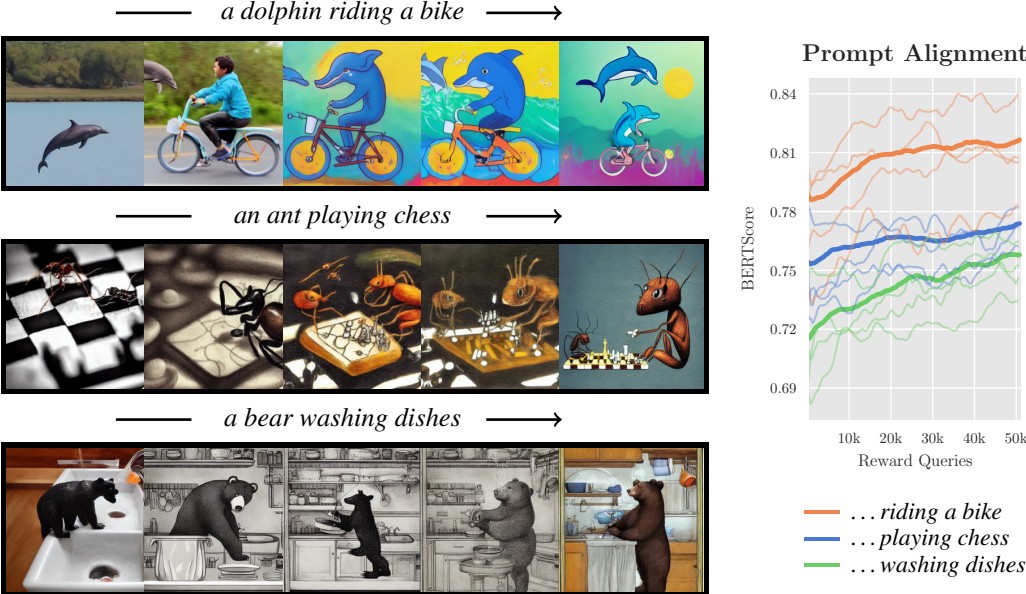

**Figure 5 (Prompt alignment) (L)** Progression of samples for the same prompt and random seed over the course of training. The images become significantly more faithful to the prompt. The samples also adopt a cartoon-like style, which we hypothesize is because the prompts are more likely depicted as illustrations than realistic photographs in the pretraining distribution. **(R)** Quantitative improvement of prompt alignment. Each thick line is the average score for an activity, while the faint lines show average scores for a few randomly selected individual prompts.

We provide a quantitative comparison of all methods in Figure 4. We plot the attained reward as a function of the number of queries to the reward function, as reward evaluation becomes the limiting factor in many practical applications. DDPO shows a clear advantage over RWR on all tasks, demonstrating that formulating the denoising process as a multi-step MDP and estimating the policy gradient directly is more effective than optimizing a reward-weighted variational bound on log-likelihood. Within the DDPO class, the importance sampling estimator slightly outperforms the score function estimator, likely due to the increased number of optimization steps. Within the RWR class, the performance of weighting schemes is comparable, making the sparse weighting scheme preferable on these tasks due to its simplicity and reduced resource requirements.

## 6.2 AUTOMATED PROMPT ALIGNMENT

We next evaluate the ability of VLMs, in conjunction with DDPO, to automatically improve the image-prompt alignment of the pretrained model without additional human labels. We focus on DDPO$_{IS}$ for this experiment, as we found it to be the most effective algorithm in Section 6.1. The prompts for this task all have the form "*a(n) [animal] [activity]*", where the animal comes from the same list of 45 common animals used in Section 6.1 and the activity is chosen from a list of 3 activities: "*riding a bike*", "*playing chess*", and "*washing dishes*".

The progression of finetuning is depicted in Figure 5. Qualitatively, the samples come to depict the prompts much more faithfully throughout the course of training. This trend is also reflected quantitatively, though is less salient as small changes in BERTScore can correspond to large differences in relevance (Zhang et al., 2020). It is important to note that some of the prompts in the finetuning set, such as "*a dolphin riding a bike*", had zero success rate from the pretrained model; if trained in isolation, this prompt would be unlikely to ever improve because there would be no reward signal. It was only via transferrable learning across prompts that these difficult prompts could improve.

Nearly all of the samples become more cartoon-like or artistic during finetuning. This was not optimized for directly. We hypothesize that this may be a function of the pretraining distribution (one would expect depictions of animals doing everyday activities to be more commonly cartoon-like than photorealistic) or of the reward function (perhaps LLaVA has an easier time recognizing the content of simple cartoon-like images).

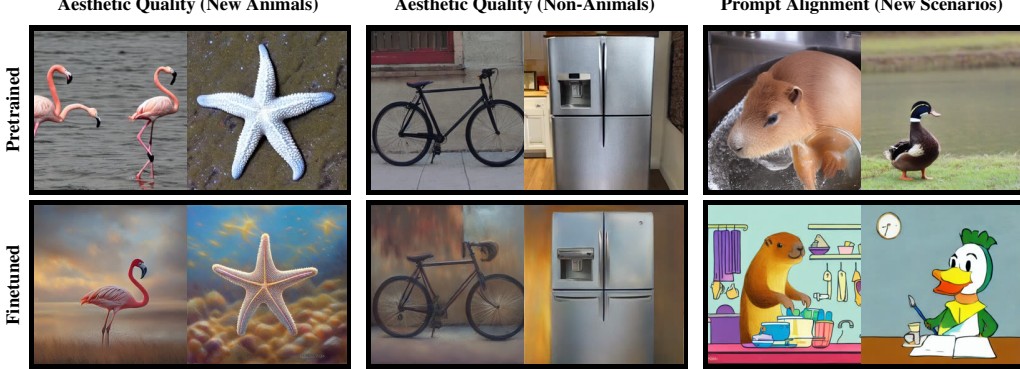

**Aesthetic Quality (New Animals)**     **Aesthetic Quality (Non-Animals)**     **Prompt Alignment (New Scenarios)**

**Figure 6 (Generalization)** Finetuning on a limited set of animals generalizes to both new animals and non-animal everyday objects. The prompts for the rightmost two columns are "*a capybara washing dishes*" and "*a duck taking an exam*". A quantitative analysis is provided in Appendix F, and more samples are provided in Appendix G.

## 6.3 GENERALIZATION

RL finetuning on large language models has been shown to produce interesting generalization properties; for example, instruction finetuning almost entirely in English has been shown to improve capabilities in other languages (Ouyang et al., 2022). It is difficult to reconcile this phenomenon with our current understanding of generalization; it would *a priori* seem more likely for finetuning to have an effect only on the finetuning prompt set or distribution. In order to investigate the same phenomenon with diffusion models, Figure 6 shows a set of DDPO-finetuned model samples corresponding to prompts that were not seen during finetuning. In concordance with instruction-following transfer in language modeling, we find that the effects of finetuning do generalize, even with prompt distributions as narrow as 45 animals and 3 activities. We find evidence of generalization to animals outside of the training distribution, to non-animal everyday objects, and in the case of prompt-image alignment, even to novel activities such as "*taking an exam*".

## 7 DISCUSSION AND LIMITATIONS

We presented an RL-based framework for training denoising diffusion models to directly optimize a variety of reward functions. By posing the iterative denoising procedure as a multi-step decision-making problem, we were able to design a class of policy gradient algorithms that are highly effective at training diffusion models. We found that DDPO was an effective optimizer for tasks that are difficult to specify via prompts, such as image compressibility, and difficult to evaluate programmatically, such as semantic alignment with prompts. To provide an automated way to derive rewards, we also proposed a method for using VLMs to provide feedback on the quality of generated images. While our evaluation considers a variety of prompts, the full range of images in our experiments was constrained (*e.g.*, animals performing activities). Future iterations could expand both the questions posed to the VLM, possibly using language models to propose relevant questions based on the prompt, as well as the diversity of the prompt distribution. We also chose not to study the problem of overoptimization, a common issue with RL finetuning in which the model diverges too far from the original distribution to be useful (see Appendix A); we highlight this as an important area for future work. We hope that this work will provide a step toward more targeted training of large generative models, where optimization via RL can produce models that are effective at achieving user-specified goals rather than simply matching an entire data distribution.

**Broader Impacts.** Generative models can be valuable productivity aids, but may also pose harm when used for disinformation, impersonation, or phishing. Our work aims to make diffusion models more useful by enabling them to optimize user-specified objectives. This adaptation has beneficial applications, such as the generation of more understandable educational material, but may also be used maliciously, in ways that we do not outline here. Work on the reliable detection of synthetic content remains important to mitigate such harms from generative models.

# 8 ACKNOWLEDGEMENTS

This work was partially supported by the Office of Naval Research and computational resource donations from Google via the TPU Research Cloud (TRC). Michael Janner was supported by a fellowship from the Open Philanthropy Project. Yilun Du and Kevin Black were supported by fellowships from the National Science Foundation.

## CODE REFERENCES

We used the following open-source libraries for this work: NumPy (Harris et al., 2020), JAX (Bradbury et al., 2018), Flax (Heek et al., 2023), optax (Babuschkin et al., 2020), h5py (Collette, 2013), transformers (Wolf et al., 2020), and diffusers (von Platen et al., 2022).

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

# APPENDIX A   OVEROPTIMIZATION

**⟵  Incompressibility  ⟶**          **Counting Animals**

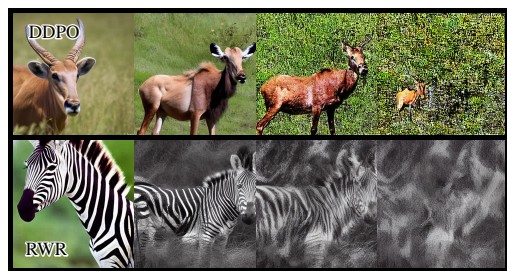
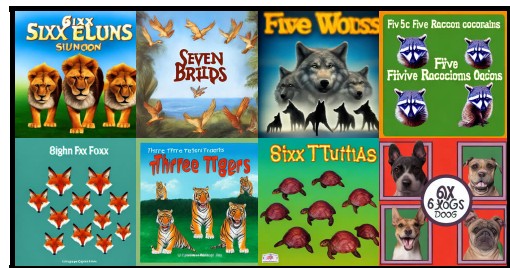

**Figure 7 (Reward model overoptimization)** Examples of RL overoptimizing reward functions. **(L)** The diffusion model eventually loses all recognizable semantic content and produces noise when optimizing for incompressibility. **(R)** When optimized for prompts of the form "*n animals*", the diffusion model exploits the VLM with a typographic attack (Goh et al., 2021), writing text that is interpreted as the specified number $n$ instead of generating the correct number of animals.

Section 6.1 highlights the optimization problem: given a reward function, how well can an RL algorithm maximize that reward? However, finetuning on a reward function, especially a learned one, has been observed to lead to reward overoptimization or exploitation (Gao et al., 2022) in which the model achieves high reward while moving too far away from the pretraining distribution to be useful.

Our setting is no exception, and we provide two examples of reward exploitation in Figure 7. When optimizing the incompressibility objective, the model eventually stops producing semantically meaningful content, degenerating into high-frequency noise. Similarly, we observed that LLaVA is susceptible to typographic attacks (Goh et al., 2021). When optimizing for alignment with respect to prompts of the form "*n animals*", DDPO exploited deficiencies in the VLM by instead generating text loosely resembling the specified number: for example, "*sixx ttutttas*" above a picture of eight turtles.

There is currently no general-purpose method for preventing overoptimization. One common strategy is to add a KL-regularization term to the reward (Ouyang et al., 2022; Stiennon et al., 2020); we refer the reader to the concurrent work of Fan et al. (2023) for a study of KL-regularization in the context of finetuning text-to-image diffusion models. However, Gao et al. (2022) suggest that existing solutions, including KL-regularization, may be empirically equivalent to early stopping. As a result, in this work, we manually identified the last checkpoint before a model began to deteriorate for each method and used that as the reference for qualitative results. We highlight this problem as an important area for future work.

# APPENDIX B   COMPARISON TO CLASSIFIER GUIDANCE

Classifier guidance (Dhariwal & Nichol, 2021) was originally introduced as a way to improve sample quality for conditional generation using the gradients from an image classifier. For a differentiable reward function such as the LAION aesthetics predictor (Schuhmann, 2022), one could naturally imagine an extension to classifier guidance that uses gradients from such a predictor to improve aesthetic score. The issue is that classifier guidance uses gradients with respect to the noisy images in the intermediate stages of the denoising process, which requires retraining the guidance network on

| Method | Aesthetic Score |
|---|---|
| Base model | $5.95 \pm 0.03$ |
| Universal guidance | $6.14 \pm 0.05$ |
| DDPO$_{\text{IS}}$ @ 20k reward queries | $6.63 \pm 0.03$ |

**Table 1** Comparison of DDPO with universal guidance using the LAION aesthetic predictor. We report the mean and one standard error over 50 samples for the prompt "wolf".

15

noisy images. Universal guidance (Bansal et al., 2023) sidesteps this issue by applying the guidance network to the fully denoised image predicted by the diffusion model at each step.

We compare DDPO with universal guidance in Table 1. We used the official implementation of universal guidance[1] with the recommended hyperparameters for style transfer, substituting the guidance network with the LAION aesthetics predictor. While universal guidance is able to produce a statistically significant improvement in aesthetic score, the change is small compared to DDPO. We only report results averaged over 50 samples for a single prompt, since universal guidance is very slow; on an NVIDIA A100 GPU, it takes almost 2 minutes to generate a single image, whereas standard generation (e.g., from a DDPO-finetuned model) takes 4 seconds.

## APPENDIX C   COMPARISON TO DPOK

Here we directly compare our implementation of DDPO to the results reported in the DPOK paper (Fan et al., 2023), which was developed concurrently with this work. The key similarities and differences between our experimental setups are summarized below:

- For this experiment only, we use Stable Diffusion v1-5 as the base model and train the UNet with low-rank adaptation (LoRA; Hu et al. (2021)) in order to match DPOK.

- Rather than matching the hyperparameters in DPOK, we use the same hyperparameters as in our other experiments (Appendix D.5) except for the learning rate which we increase to 3e-4. We found that when using LoRA, a higher learning rate is necessary to get comparable performance to full finetuning.

- Like DPOK, we train on four prompts: "a green colored rabbit" (color), "four wolves in the park" (count), "a dog and a cat" (composition), and "a dog on the moon" (location). Unlike DPOK, we train a single model for all four prompts.

- Like DPOK, we train the model using ImageReward (Xu et al., 2023) as the reward function. We evaluate the model using ImageReward and the LAION aesthetics predictor (Schuhmann, 2022).

- Unlike DPOK, we do not employ KL regularization.

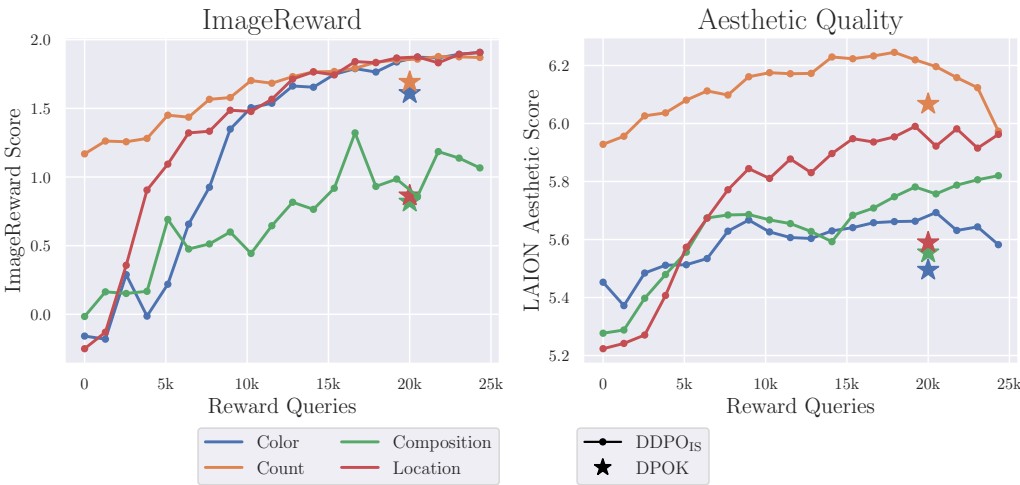

**Figure 8** Comparison of DDPO$_{\text{IS}}$ with DPOK. We take the DPOK numbers directly from the paper, which only reports scores at one point in training (after 20k reward queries). Like in DPOK, scores are averaged over 50 samples for each prompt.

[1] https://github.com/arpitbansal297/Universal-Guided-Diffusion

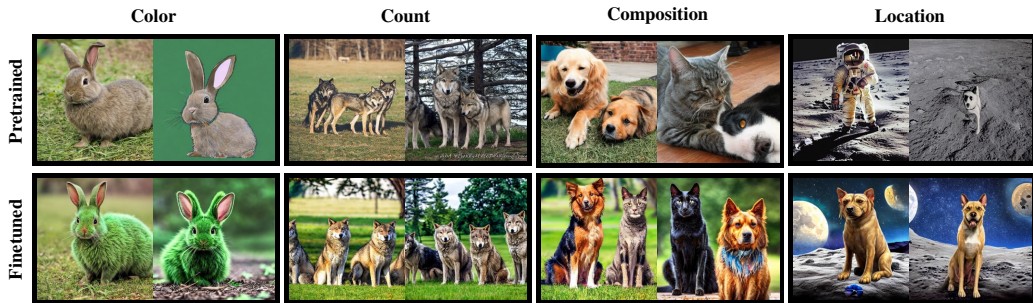

**Figure 9** Qualtitative examples of the results of ImageReward training on the DPOK prompts: "a green colored rabbit" (color), "four wolves in the park" (count), "a dog and a cat" (composition), and "a dog on the moon" (location). The finetuned images are generated from a model trained for 20k reward queries.

The results are presented in Figure 8. Our implementation of DDPO$_{\text{IS}}$ outperforms DPOK accross the board, without using KL regularization. Figure 8 also doubles as a quantitative study of overoptimization (Appendix A), since the model is trained with one reward function (ImageReward) and evaluated with another (LAION aesthetic score). We find that significant overoptimization does begin to happen within 25k reward queries for one of the prompts (count: "four wolves in the park"), which is reflected by a drop in LAION aesthetic score. However, the overoptimization is not severe or unreasonably fast. We provide qualitative samples in Figure 9 showing that the model is able to produce high-quality images at 20k reward queries.

## APPENDIX D   IMPLEMENTATION DETAILS

For all experiments, we use Stable Diffusion v1.4 (Rombach et al., 2022) as the base model and finetune only the UNet weights while keeping the text encoder and autoencoder weights frozen.

### D.1   DDPO IMPLEMENTATION

We collect 256 samples per training iteration. For DDPO$_{\text{SF}}$, we accumulate gradients across all 256 samples and perform one gradient update. For DDPO$_{\text{IS}}$, we split the samples into 4 minibatches and perform 4 gradient updates. Gradients are always accumulated across all denoising timesteps for a single sample. For DDPO$_{\text{IS}}$, we use the same clipped surrogate objective as in proximal policy optimization (Schulman et al., 2017), but find that we need to use a very small clip range compared to standard RL tasks. We use a clip range of 1e-4 for all experiments.

### D.2   RWR IMPLEMENTATION

We compute the weights for a training iteration using the entire dataset of samples collected for that training iteration. For $w_{\text{RWR}}$, the weights are computed using the softmax function. For $w_{\text{sparse}}$, we use a percentile-based threshold, meaning $C$ is dynamically selected such that the bottom $p\%$ of a given pool of samples are discarded and the rest are used for training.

### D.3   REWARD NORMALIZATION

In practice, rewards are rarely used as-is, but instead are normalized to have zero mean and unit variance. Furthermore, this normalization can depend on the current state; in the policy gradient context, this is analogous to a value function baseline (Sutton et al., 1999), and in the RWR context, this is analogous to advantage-weighted regression (Peng et al., 2019). In our experiments, we normalize the rewards on a per-context basis. For DDPO, this is implemented as normalization by a running mean and standard deviation that is tracked for each prompt independently. For RWR, this is implemented by computing the softmax over rewards for each prompt independently. For RWR$_{\text{sparse}}$, this is implemented by computing the percentile-based threshold $C$ for each prompt independently.

## D.4 Resource Details

RWR experiments were conducted on a v3-128 TPU pod, and took approximately 4 hours to reach 50k samples. DDPO experiments were conducted on a v4-64 TPU pod, and took approximately 4 hours to reach 50k samples. For the VLM-based reward function, LLaVA inference was conducted on a DGX machine with 8 80Gb A100 GPUs.

## D.5 Full Hyperparameters

|  |  | $\text{DDPO}_{\text{IS}}$ | $\text{DDPO}_{\text{SF}}$ | RWR | $\text{RWR}_{\text{sparse}}$ |
|---|---|---|---|---|---|
| Diffusion | Denoising steps ($T$) | 50 | 50 | 50 | 50 |
|  | Guidance weight ($w$) | 5.0 | 5.0 | 5.0 | 5.0 |
| Optimization | Optimizer | AdamW | AdamW | AdamW | AdamW |
|  | Learning rate | 1e-5 | 1e-5 | 1e-5 | 1e-5 |
|  | Weight decay | 1e-4 | 1e-4 | 1e-4 | 1e-4 |
|  | $\beta_1$ | 0.9 | 0.9 | 0.9 | 0.9 |
|  | $\beta_2$ | 0.999 | 0.999 | 0.999 | 0.999 |
|  | $\epsilon$ | 1e-8 | 1e-8 | 1e-8 | 1e-8 |
|  | Gradient clip norm | 1.0 | 1.0 | 1.0 | 1.0 |
| RWR | Inverse temperature ($\beta$) | - | - | 0.2 | - |
|  | Percentile | - | - | - | 0.9 |
|  | Batch size | - | - | 128 | 128 |
|  | Gradient updates per iteration | - | - | 400 | 400 |
|  | Samples per iteration | - | - | 10k | 10k |
| DDPO | Batch size | 64 | 256 | - | - |
|  | Samples per iteration | 256 | 256 | - | - |
|  | Gradient updates per iteration | 4 | 1 | - | - |
|  | Clip range | 1e-4 | - | - | - |

## D.6 List of 45 Common Animals

This list was used for experiments with the aesthetic quality reward function and the VLM-based reward function.

| | | | | | | | | |
|---|---|---|---|---|---|---|---|---|
| cat | dog | horse | monkey | rabbit | zebra | spider | bird | sheep |
| deer | cow | goat | lion | tiger | bear | raccoon | fox | wolf |
| lizard | beetle | ant | butterfly | fish | shark | whale | dolphin | squirrel |
| mouse | rat | snake | turtle | frog | chicken | duck | goose | bee |
| pig | turkey | fly | llama | camel | bat | gorilla | hedgehog | kangaroo |

# Appendix E  Additional Design Decisions

## E.1 CFG Training

Recent text-to-image diffusion models rely critically on *classifier-free guidance* (CFG) (Ho & Salimans, 2021) to produce perceptually high-quality results. CFG involves jointly training the diffusion model on conditional and unconditional objectives by randomly masking out the context $\mathbf{c}$ during training. The conditional and unconditional predictions are then mixed at sampling time using a guidance weight $w$:

$$\tilde{\epsilon}_\theta(\mathbf{x}_t, t, \mathbf{c}) = w\epsilon_\theta(\mathbf{x}_t, t, \mathbf{c}) + (1 - w)\epsilon_\theta(\mathbf{x}_t, t) \tag{3}$$

where $\epsilon_\theta$ is the $\epsilon$-prediction parameterization of the diffusion model (Ho et al., 2020) and $\tilde{\epsilon}_\theta$ is the guided $\epsilon$-prediction that is used to compute the next denoised sample.

For reinforcement learning, it does not make sense to train on the unconditional objective since the reward may depend on the context. However, we found that when only training on the conditional objective, performance rapidly deteriorated after the first round of finetuning. We hypothesized that

this is due to the guidance weight becoming miscalibrated each time the model is updated, leading to degraded samples, which in turn impair the next round of finetuning, and so on. Our solution was to choose a fixed guidance weight and use the guided $\epsilon$-prediction during training as well as sampling. We call this procedure *CFG training*. Figure 10 shows the effect of CFG training on RWR$_{\text{sparse}}$; it has no effect after a single round of finetuning, but becomes essential for subsequent rounds.

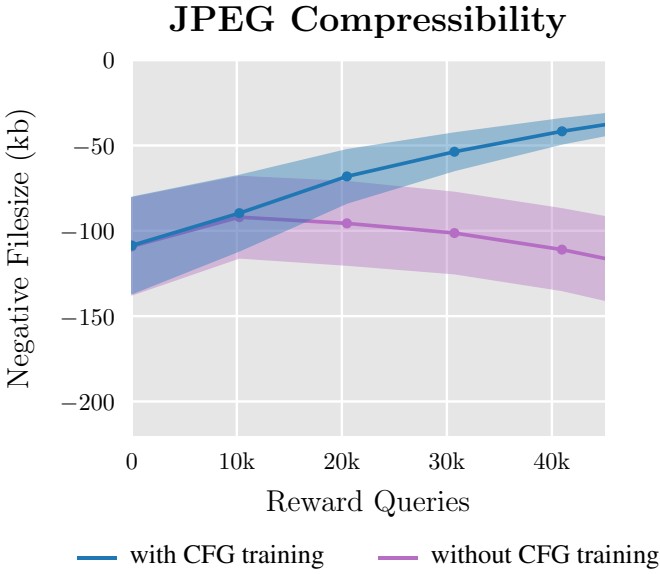

**Figure 10 (CFG training)** We run the RWR$_{\text{sparse}}$ algorithm while optimizing only the conditional $\epsilon$-prediction (*without CFG training*), and while optimizing the guided $\epsilon$-prediction (*with CFG training*). Each point denotes a diffusion model update. We find that CFG training is essential for methods that do more than one round of interleaved sampling and training.

### E.2 INTERLEAVING

There are two main differences between DDPO and RWR, as compared in Section 6.1: the objective (DDPO uses the policy gradient) and the data distribution (DDPO is significantly more on-policy, collecting 256 samples per iteration as opposed to 10,000 for RWR). This choice is motivated by standard RL practice, in which policy gradient methods specifically require on-policy data (Sutton et al., 1999), whereas RWR is designed to work in on off-policy data (Nair et al., 2020) and is known to underperform other algorithms in more online settings (Duan et al., 2016).

However, we can isolate the effect of the data distribution by varying how interleaved the sampling and training are in RWR. At one extreme is a single-round algorithm (Lee et al., 2023), in which $N$ samples are collected from the pretrained model and used for finetuning. It is also possible to run $k$ rounds of finetuning each on $\frac{N}{k}$ samples collected from the most up-to-date model. In Figure 11, we evaluate this hyperparameter and find that increased interleaving does help up to a point, after which it causes performance degradation. However, RWR is still unable to match the asymptotic performance of DDPO at any level of interleaving.

### APPENDIX F  QUANTITATIVE RESULTS FOR GENERALIZATION

In Section 6.3, we presented qualitative evidence of both the aesthetic quality model and the image-prompt alignment model generalizing to prompts that were unseen during finetuning. In Figure 12, we provide an additional quantitative analysis of generalization with the aesthetic quality model, where we measure the average reward throughout training for several prompt distributions. In accordance with the qualitative evidence, we see that the model generalizes very well to unseen animals, and everyday objects to a lesser degree.

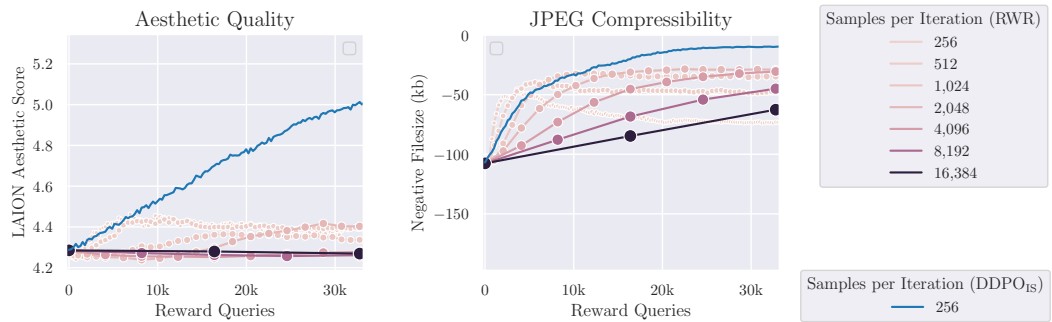

**Figure 11 (RWR interleaving ablation)** Ablation over the number of samples collected per iteration for RWR. The number of gradient updates per iteration remains the same throughout. We find that more frequent interleaving is beneficial up to a point, after which it causes performance degradation. However, RWR is still unable to match the asymptotic performance of DDPO at any level of interleaving.

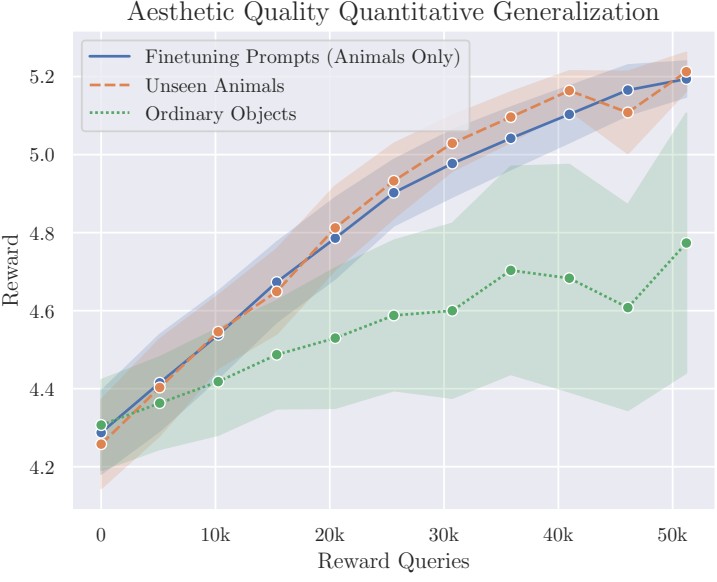

**Figure 12 (Quantitative generalization)** Reward curves demonstrating the generalization of the aesthetic quality objective to prompts not seen during finetuning. The finetuning prompts are a list of 45 common animals, "unseen animals" is a list of 38 additional animals, and "ordinary objects" is a list of 50 objects (e.g. toaster, chair, coffee cup, etc.).

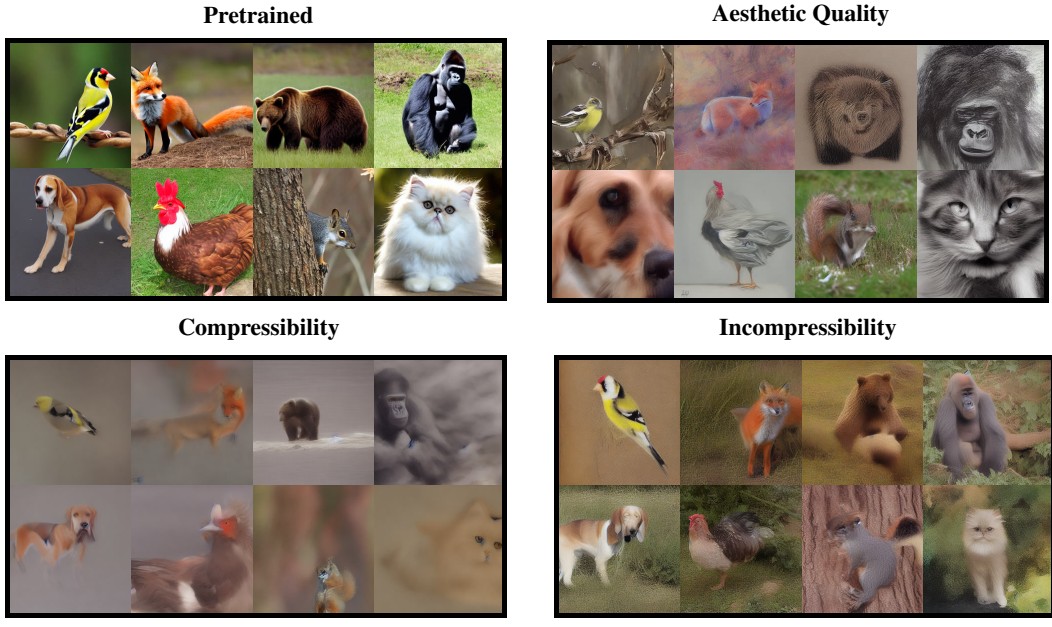

**Figure 13 (RWR samples)**

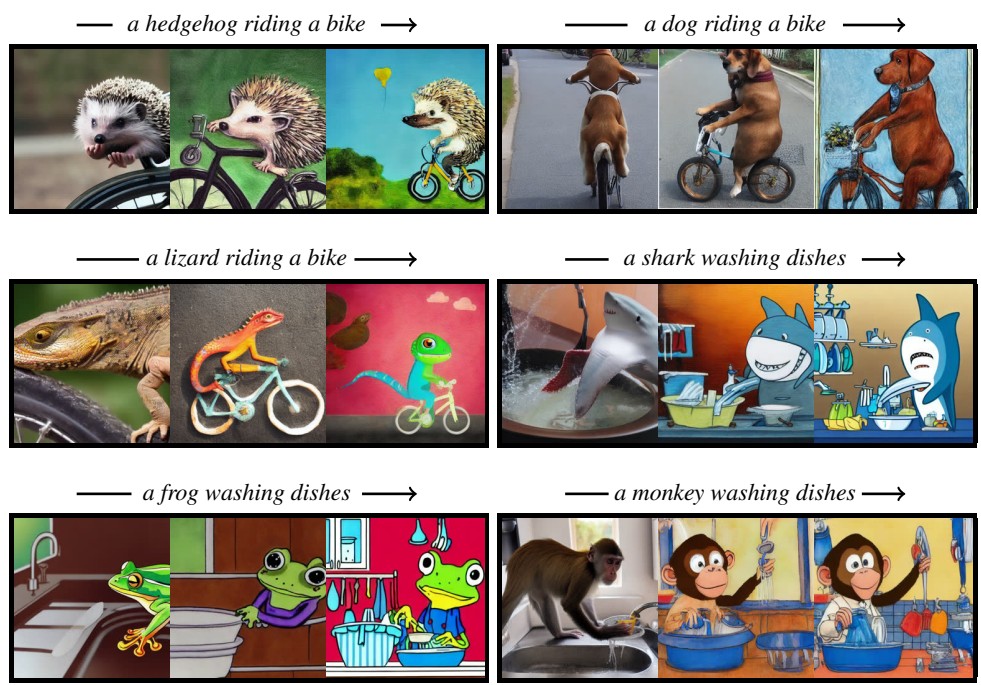

**Figure 14 (More image-prompt alignment samples)**

## APPENDIX G   MORE SAMPLES

Figure 13 shows qualitative samples from the baseline RWR method. Figure 14 shows more samples on seen prompts from DDPO finetuning with the image-prompt alignment reward function. Figure 15 shows more examples of generalization to unseen animals and everyday objects with the aesthetic quality reward function. Figure 16 shows more examples of generalization to unseen subjects and activities with the image-prompt alignment reward function.

**Pretrained (New Animals)**        **Aesthetic Quality (New Animals)**

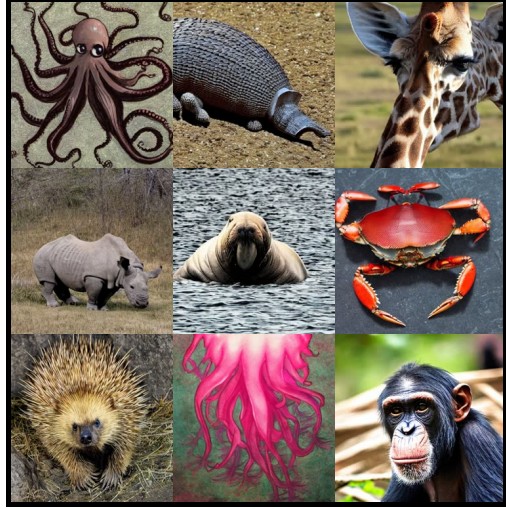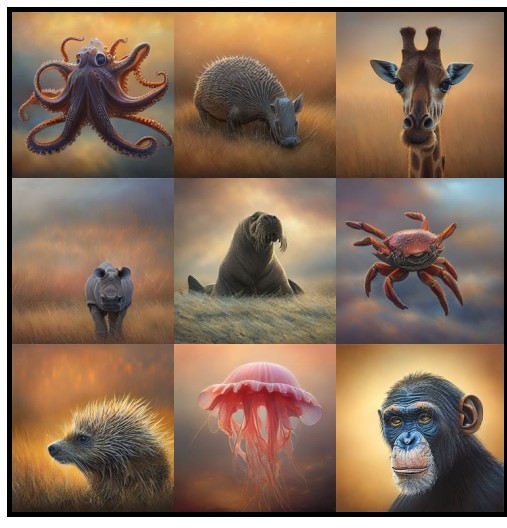

**Pretrained (Non-Animals)**        **Aesthetic Quality (Non-Animals)**

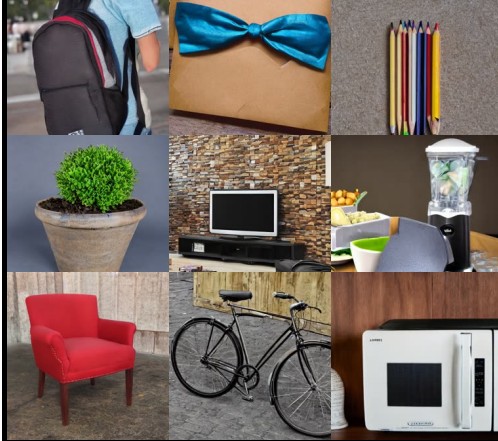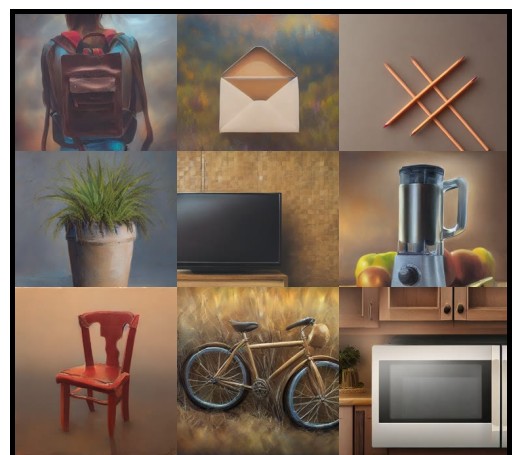

**Figure 15 (Aesthetic quality generalization)**

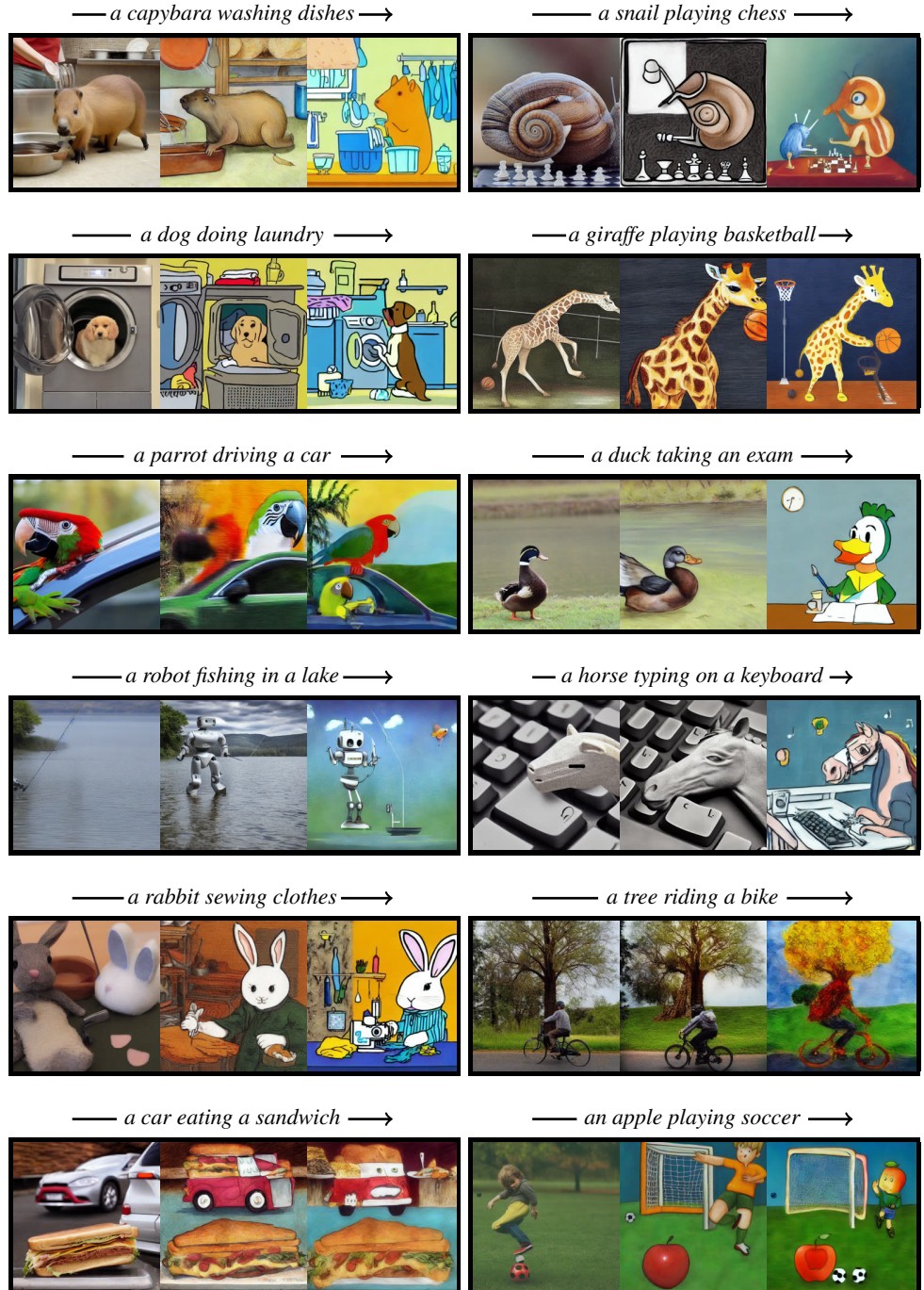

**Figure 16 (Image-prompt alignment generalization)**

