# OpenReview forum: "Training Diffusion Models with Reinforcement Learning"
_ICLR.cc/2024/Conference — ICLR 2024 poster_

### Official Review · Reviewer_rYFt · 2023-10-31

**Soundness:** 2 fair
**Presentation:** 3 good
**Contribution:** 2 fair
**Rating:** 6
**Confidence:** 3

**Summary:**

In this paper, the authors propose a policy gradient algorithm to finetune text-to-image diffusion models for downstream tasks using only a reward function. Specifically, they reframe the denoising process as a multi-step MDP and introduce two variants of denosing diffusion policy optimization. In addition, they validate the proposed method on different downstream tasks, such as aesthetics and prompt-image alignment. In particular, they propose a automated VLM-based reward function for prompt-image alignment.

**Strengths:**

+ The proposed method is effective and validated with various experiments. Besides, this paper is simple and easy to implement.
+ The proposed automated prompt alignment is somewhat practical and provide an alternative for prompt-image alignment.

**Weaknesses:**

- The contribution of the proposed method is minor, though it is effective on different downstream tasks. Compared with DPOK, the modification of policy optimization is somewhat incremental. In the related work, the authors claims that training on many prompts is one of two key advantages over DPOK. But, DPOK also can train with multiple prompts and provides quantitative results for that experiment. It is better to conduct this experiment with the same setting to support this point. Besides, the authors only show that the proposed method outperforms simple RWR methods. It is essential to compare with previous works. For example, show different methods’ ability to handle prompts involving color, composition, counting and location.
-  As shown in Appendix B, there is a subtle bug in the implementation. This bug maybe affect the quantitative comparisons and lower the support for the effectiveness of the proposed method. It is one of my major concerns about this paper.
- In the experiment related to aesthetic quality, the finetuned model produce images with fixed style, as shown in Figure 3 and Figure 6. Is it only attributed to the bug in the implementation? Based on this phenomenon, I raise a concern: does the proposed method compromise the diversity of text-to-image diffusion models?

**Questions:**

- In the related work, ‘Reward-Weighted Regression’ should be used before using its abbreviation (RWR). If not, it maybe a misleading phrase for the readers, especially ones without context.

---

> ### Author Response · Authors · 2023-11-18
> **Rebuttal + new experiments**
>
> Thank you for your thorough review. It seems that your primary concerns are novelty with respect to DPOK and the subtle bug affecting the aesthetic score. Regarding the bug, we have re-run the primary quantitative comparison with the fixed reward function. Regarding DPOK, as we explain in the top-level rebuttal, our paper was posted to arXiv 3 days before DPOK, and as such the work should be considered concurrent.
>
> That said, we have added an experiment directly comparing our method to DPOK in Appendix C of the updated PDF ([screenshot of quantitative results;](https://ibb.co/Jc0LmS0) [screenshot of qualitative results](https://ibb.co/z4shhBd)). Our method significantly outperforms the numbers reported by DPOK on the same task, even without KL regularization.
>
> > **As shown in Appendix B, there is a subtle bug in the implementation. This bug maybe affect the quantitative comparisons and lower the support for the effectiveness of the proposed method. It is one of my major concerns about this paper.**
>
> Apologies for any confusion, but this bug only affected the leftmost chart in Figure 4 (primary comparison) and the leftmost chart in the RWR interleaving ablation in the appendix. No other results, qualitative or quantitative, were affected. While the affected results still represented a fair scientific comparison, we decided to mention the bug in the appendix in the interest of full transparency.
>
> We have re-run the primary comparison in Figure 4 and updated the submission. Unfortunately, it is infeasible to re-run the extra interleaving baseline/ablation in the appendix due to compute limitations. However, the leftmost chart still provides a fair comparison, since all methods were tested on the same reward function, and the rightmost chart supports the same conclusion while being unaffected by the bug.
>
> > **does the proposed method compromise the diversity of text-to-image diffusion models**
>
> Yes, we do notice convergence on a similar style for the aesthetic quality experiments. A similar phenomenon has been observed with RLHF for language models ([Kirk et. al., 2023](https://arxiv.org/abs/2310.06452)). While reduction of diversity is a concern, we do not believe that this is fatal to the practical applicability or intellectual contribution of our method, and could be alleviated in future work. We will be sure to add a discussion of sample diversity to the limitations section for the camera-ready.
>
> > **In the related work, ‘Reward-Weighted Regression’ should be used before using its abbreviation (RWR).**
>
> Thanks for the catch! We have fixed this in the updated submission.

---

> > ### Comment · Reviewer_rYFt · 2023-11-22
> >
> > I have read all reviews and authors’ rebuttal. The authors resolve my concerns. Thus, I raise my score to 6.

---

> ### Author Response · Authors · 2023-11-21
>
> Hi reviewer rYFt,
>
> We hope we have addressed all of your stated weaknesses, as well as answered your questions about the paper. Most notably, we have clarified that DPOK is concurrent work and thus the DDPO algorithm should be considered an original intellectual contribution. We have also fixed the issue with the aesthetic predictor, which was one of your major concerns.
>
> As such, we would greatly appreciate it if you could take a look at our rebuttal and let us know if you still think our paper is below the acceptance threshold for ICLR.
>
> Regards,
> The Authors

---

### Official Review · Reviewer_EgHv · 2023-10-31

**Soundness:** 2 fair
**Presentation:** 3 good
**Contribution:** 2 fair
**Rating:** 6
**Confidence:** 4

**Summary:**

The paper introduces a method that poses the diffusion process as an MDP and applies reinforcement learning to text-to-image diffusion models. It uses different reward functions, such as image compressibility, prompt alignment and image quality. The authors formulate three different approaches to fine-tune image diffusion models, two involving multi-step reinforcement learning and one that re-weights the original inverse diffusion objective. The authors evaluate their method on relevant datasets and discuss relevant shortcomings.

**Strengths:**

- the paper proposes a novel work (other work mentioned in related works section is concurrent work as it will only be published at Neurips in December)
- the approach is simple yet effective
- a variety of reward functions are explored and all yield visually pleasing results

**Weaknesses:**

- The proposed method does not consider the problem of overoptimisation, instead the authors argue that early stopping from visual inspection is sufficient. This makes the method applicable to problems where visual inspection is possible (which is likely the case for many image tasks). However, it renders the method inapplicable to problems where the needed visual inspection is not possible. (E.g. one might not be able to apply this method to a medical imaging task where visual inspection by a human supervisor does not allow to determine when to stop the optimisation process).
- Further, the reliance on visual inspection negatively impacts scalability of the method.
- The evaluation is missing a simple classifier guidance baseline, where samples are generated from the diffusion model with classifier guidance according to the given reward function. (Classifier guidance is mentioned in the related work but not evaluated as a baseline)
 - The hypothesis that cartoon-like samples are more likely after fine-tuning because no photorealistic images exist for the given prompt category could be verified by fine-tuning on categories for which photo-realistic images do exist .
- the generalization study in 6.3. is purely qualitative, a quantitative analysis would be more adequate
- similarly, overoptimization could be quantitatively analysed

**Questions:**

- I am wondering why the lacking quantitative experiments have not been conducted (or why they cannot be conducted)
- Similarly, I am wondering how the authors think about the early stopping mechanism

---

> ### Author Response · Authors · 2023-11-18
> **Rebuttal + new experiments**
>
> Thank you for your thorough review. It seems that your primary concern is about overoptimization. It’s worth noting that we found it reasonably straightforward to run our method for a fixed number of iterations (e.g., 15k reward queries) and recover good generations; we also did not observe completely degenerated images in the VLM or aesthetic quality experiments, only in compressibility and incompressibility (when run for long enough). We have added an additional quantitative experiment that we hope addresses your concern, which we discuss more below.
>
> We would also like to point out that overoptimization is an open research problem that is well outside the scope of this paper. While we agree that it is certainly a major issue that hinders many real-life applications (such as the provided medical imaging example), the issue of overoptimization also exists in language modeling to an equivalent degree ([Gao et. al., 2022](https://arxiv.org/abs/2210.10760)), yet it has certainly not prevented RLHF from having a huge impact in that domain ([Ouyang et. al., 2022](https://arxiv.org/abs/2203.02155); [Bai et. al., 2022](https://arxiv.org/abs/2212.08073)). Current techniques for combating overoptimization, such as KL-regularization or automated evaluation, do not completely eliminate the need for human evaluation in real-life systems.
>
> > **the generalization study in 6.3. is purely qualitative, a quantitative analysis would be more adequate**
>
> A quantitative analysis of generalization was already provided in Appendix D, which is referenced in the caption of Figure 6.
>
> > **overoptimization could be quantitatively analysed**
>
> We have provided a quantitative analysis of overoptimization in Appendix C of the updated submission ([screenshot](https://ibb.co/Jc0LmS0)). We find that both the optimized metric (ImageReward) and held-out metric (Aesthetic Quality) increase for quite some time, during which the model produces high-quality generations ([screenshot](https://ibb.co/z4shhBd)). Overoptimization — reflected by a decrease in the held-out metric — does begin to occur for one prompt near the end of training, but even then it is not fatal. Such held-out metrics could also be used to automatically decide a stopping point.
>
> > **The evaluation is missing a simple classifier guidance baseline**
>
> We have added a comparison to classifier guidance in Appendix B of the updated submission ([screenshot](https://ibb.co/dfJbVTw)). We note that only the aesthetic predictor reward function is differentiable, meaning it is the only one where classifier guidance is applicable. See the top-level rebuttal for more information.

---

> ### Author Response · Authors · 2023-11-21
>
> Hi reviewer EgHv,
>
> We hope we have addressed most of your weaknesses, as well as answered your questions about our paper. To be specific, we have added the requested quantitative evaluation of overoptimization and a new classifier guidance baseline, as well as provided some clarification about overoptimization, which seemed to be your main concern. As such, we would greatly appreciate it if you could take a look at our rebuttal and let us know if you still think our paper is below the acceptance threshold for ICLR.
>
> Thanks,
> The Authors

---

> > ### Comment · Reviewer_EgHv · 2023-11-21
> >
> > Dear authors, thank you for your thorough rebuttal, which addressed most of my concerns and answered most of my questions.
> >
> > I am still curious about testing the hypothesis that cartoon-like samples are more likely after fine-tuning because no photorealistic images exist for the given prompt category by fine-tuning on categories for which photo-realistic images do exist. Whats your take on this?

---

> > > ### Comment · Reviewer_EgHv · 2023-11-21
> > >
> > > I agree with the authors that DPOK is concurrent work. Hence it should not be used for assessing this paper in any way.

---

> > > ### Author Response · Authors · 2023-11-21
> > >
> > > Thanks for the follow-up! We have not had a chance to run more experiments with the VLM reward function, but the additional experiment we added in Appendix C is potentially related to your question. In that experiment, we use ImageReward --- another prompt-image alignment objective --- on a different set of prompts, such as "a green colored rabbit" and "a dog on the moon". Based on the qualitative results ([screenshot](https://ibb.co/z4shhBd)), the model is able to produce well-aligned photorealistic images for these prompts. While we think this is likely due to the difference in prompts, the effect of the prompts and reward function could be disentangled by also running the ImageReward experiment with the "animals doing activities" prompts. Unfortunately, that experiment would not finish before the end of the discussion period.

---

> > > > ### Comment · Reviewer_EgHv · 2023-11-22
> > > >
> > > > Hi, thanks for the response. In all fairness, I wanted to point out that this is not a new request that I just brought up a day before the rebuttal deadline, but that I had written exactly this in my initial review of the paper.

---

> > > > > ### Author Response · Authors · 2023-11-22
> > > > >
> > > > > Sorry, yes we regret that we couldn't get to your question sooner. It's a good suggestion that would definitely provide some good insights. Unfortunately we had to prioritize and were busy running quite a few other experiments for this rebuttal. We actually did start the aforementioned run yesterday, and it seems that the load on our machine is a bit less today, so it may finish in time after all. We'll post an update tonight if it has progressed enough to get some interesting results.

---

> > > > > ### Author Response · Authors · 2023-11-23
> > > > >
> > > > > Hi reviewer EgHv, unfortunately we haven't been able to get the results. For now, we'll remove the statement from the paper that hypothesizes why the images become cartoon-like, instead saying we don't know why. We'll run some more experiments to include in the appendix for the camera-ready, and update the main paper if we're able to answer the question conclusively.

---

### Official Review · Reviewer_sWSo · 2023-10-31

**Soundness:** 4 excellent
**Presentation:** 4 excellent
**Contribution:** 3 good
**Rating:** 8
**Confidence:** 4

**Summary:**

This paper introduces Denoising Diffusion Policy Optimization (DDPO), where they fine-tune diffusion models by reinforcement learning (RL). In specific, given the scalar reward function that takes an image and conditions (e.g., text prompts), DDPO considers the reverse generative process as Markovian Decision Process (MDP), where the reward is only given at the zeroth timestep. The authors implement DDPO via policy gradient (i.e., REINFORCE) or together with importance sampling (i.e., PPO-like approach). Through experiments they show that DDPO can be used in improving aesthetic quality, image-text alignment, and compression (or incompression) of an image, and verify its effectiveness in comparison to reward augmented fine-tuning.

**Strengths:**

- This paper is clearly written and easy to follow.
- The problem of solving diffusion generative processes as solving MDP is clearly stated, and the proposed method generalizes prior works (i.e., reward-weighted regression (RWR)) for the multi-step MDP case.
- The experiments clearly validate the efficiency of DDPO over prior diffusion model tuning with reward functions as well as detailed algorithmic choices are provided.

**Weaknesses:**

- After RL fine-tuning, the generated images seem to be saturated. For example, the fine-tuned models generate images with high aesthetic scores, but they seem to generate images with similar backgrounds of sunset. For prompt alignment experiments, the models generate cartoon-like images.
- I think one of the main contributions of the paper is on utilizing VLMs for optimizing text-to-image diffusion models. In this context, the discussion on the choice of reward function should be discussed more in detail. For example, I think different instruction prompts for the LLaVA model would make different results, but the paper lacks details in the choice of reward function.

**Questions:**

- Regarding the capability of DDPO, is it possible to control the image generation to have predicted reward? For example, Classifier-Free guidance provides control over sample fidelity and diversity. It seems like DDPO improves prompt fidelity at the expense of sample diversity, so I wonder if there is a way to control the amount of reward during generation.
- It seems like the reward function for different tasks have different ranges, and thus DDPO uses running mean and variance for normalization. I guess the difficulty of solving MDP varies across different prompts. Could the author elaborate on the optimization analysis with respect to the difficulty of prompts?
- In implementation, the author chooses T=50 throughout experiments. What if the timesteps become smaller or larger? Definitely the denser timestep sampling would require more cost, but I believe it can reduce the variance of gradient. Could the author elaborate on this point?
- Do the authors try different reward models for image-text alignment? How robust is DDPO algorithm with respect to different reward functions?

---

> ### Author Response · Authors · 2023-11-18
> **Rebuttal + new experiments**
>
> Thank you for your thoughtful review. It seems that your primary concerns are the saturation of images during finetuning and the lack of discussion of VLM reward functions. Regarding the first point, we would like to point out that this is a well-known phenomenon that also affects RL finetuning of language models ([Kirk et. al., 2023](https://arxiv.org/abs/2310.06452)), yet it has certainly not prevented RLHF from having a huge impact in that domain. Regarding the second point, we would like to emphasize that the primary intellectual contribution of this paper is the algorithm itself, and we believe the thorough analysis of 4 reward functions in the main paper meets the bar for ICLR acceptance. That said, we have added an additional experiment that uses ImageReward in Appendix C ([screenshot](https://ibb.co/Jc0LmS0)).
>
> > **After RL fine-tuning, the generated images seem to be saturated**
>
> Yes, we do notice convergence on a similar style for the aesthetic quality experiments. A similar phenomenon has been observed with RLHF for language models ([Kirk et. al., 2023](https://arxiv.org/abs/2310.06452)). While reduction of diversity is a concern, we do not believe that this is fatal to the practical applicability or intellectual contribution of our method, and could be alleviated in future work. We will be sure to add a discussion of sample diversity to the limitations sections for the camera-ready.
>
> > **the discussion on the choice of VLM reward function should be discussed more in detail**
>
> We briefly discuss the choice of LLaVA prompt in the third paragraph of Section 5.3, where we mention that the design space of VLM reward functions is virtually unlimited and that we look forward to future work that explores this direction. In the camera-ready, we will elaborate on why we chose LLaVA (it was the best open-source VLM at the time) and the evaluation scheme that we did (we chose to stick with the simplest, most general prompt, but found that BERTScore was necessary to increase reward granularity).
>
> > **is it possible to control the image generation to have predicted reward?**
>
> The most direct way would be early stopping — earlier checkpoints in the training process have lower reward, in exchange for being closer to the distribution of the pretrained model.
>
> > **Could the author elaborate on the optimization analysis with respect to the difficulty of prompts?**
>
> Some prompts may be naturally more difficult for the pretrained model than others. For example, the  model may easily generate “a man riding a horse,” but struggle to generate well-aligned images for “a horse riding a man.” Figure 5 provides a quantitative example of this: the “riding a bike” prompts achieve the highest average reward, followed by “playing chess,” and then “washing dishes.”
>
> > **In implementation, the author chooses T=50 throughout experiments. What if the timesteps become smaller or larger?**
>
> Using more generation timesteps would certainly improve policy gradient estimates, as well as improve the base generation quality of the model. 50 timesteps is a standard choice in the literature, and we found it to work well in our experiments, so we did not ablate this hyperparameter. We suspect that one would find diminishing returns if the number of timesteps was increased much beyond 50.
>
> > **Do the authors try different reward models for image-text alignment?**
>
> We have included an additional experiment with ImageReward in Appendix C ([screenshot](https://ibb.co/Jc0LmS0)). This and LLaVA are the only two models for image-text alignment that we have personally tried.

---

> ### Author Response · Authors · 2023-11-21
>
> Hi reviewer sWSo,
>
> We hope we have addressed both of your stated weaknesses, as well as answered all of your questions. As such, we would greatly appreciate it if you could take a look at our rebuttal, letting us know if you find the responses adequate, and if so, if you would consider raising your score.
>
> Thanks,
> The Authors

---

> > ### Comment · Reviewer_sWSo · 2023-11-21
> > **Response**
> >
> > Thanks for your response, most of my concerns are addressed. I believe this is a primary work on fine-tuning the diffusion model with reward models (with concurrent work DPOK). However, as other reviewers also noted, DDPO might cause the saturation problem and the choice of reward function (e.g., LLaVA score) could be better elaborated. I am generally positive about accepting this paper. I update my score to 8.

---

### Official Review · Reviewer_qqX7 · 2023-11-03

**Soundness:** 3 good
**Presentation:** 2 fair
**Contribution:** 1 poor
**Rating:** 5
**Confidence:** 4

**Summary:**

This paper presents a method for fine-tuning large-scale diffusion-based text generation models using reinforcement learning. It views the reverse process of diffusion models as a reinforcement learning process with time steps T (T is the denoising steps), where the output of the diffusion model at each step serves both as an action and the observation for the next moment. The authors describe how posing denoising as a multi-step decision-making problem enables a class of policy gradient algorithms, referred to as denoising diffusion policy optimization. And the authors also employ an open-source image quality evaluator as the reward function and also devise an automated prompt-image alignment method using feedback from a vision-language model.

**Strengths:**

This paper is well-written, with clear logic and beautifully crafted figures, making it easy to follow the authors' line of reasoning. Additionally, the paper is well-structured, presenting an easy-to-follow approach to fine-tuning diffusion models using reinforcement learning for alignment. The narrative is straightforward, and the methods described are, in my opinion, sensible.

**Weaknesses:**

1) In terms of image generation quality, the paper lacks a quantitative and qualitative comparison with recent works. It fails to provide experimental support for its effectiveness. Specifically, in the absence of comparisons in image quality with all methods related to "Optimizing diffusion models using policy gradients," it is challenging to discern the improvements this paper offers over baseline approaches. This makes it difficult to evaluate the paper's contribution to the community.

2) Regarding originality and community contribution, compared to DPOK and "Optimizing ddpm sampling with shortcut fine-tuning," I did not observe significant differences between this paper and DPOK from the introduction to the Method section. The structural similarity in writing is quite evident. The most noticeable difference is the introduction of importance sampling in method 4.3. However, this alone does not sufficiently support the paper’s claims of innovation, especially without experimental evidence backing its effectiveness (if such evidence exists, please inform me during the Rebuttal phase). The second noticeable difference is the automated generation of rewards using BLIP, which is already a standard engineering practice and has been claimed by RLAIF. I do not believe the methodological contributions and community impact of this paper meet the acceptance standards of the ICLR conference.

3) The experimental evaluation criteria are unreasonable. Most pretrained reward evaluation models are not designed for robust data manifold, especially for treating it as a "fine-tuning teacher." Consequently, out-of-domain evaluation is necessarily needed. For example, the authors use Aesthetic Quality as a reward function to fine-tune the Diffusion model and employ the same Aesthetic Quality for scoring during evaluation. This approach does not allow for assessing whether an improvement in the Aesthetic Quality Score correlates with an enhancement in image generation quality. A more reasonable evaluation, as exemplified by DPOK, would involve fine-tuning with one Reward model and then evaluating it against both this model and a new, out-of-domain Reward model. DPOK's evaluating approach might provide a more substantial basis for assessment.

4) Lack of code. The lack of open-source code, compounded by the absence of comparisons with recent works, makes it difficult to assess the practical feasibility of the approach.

**Questions:**

My main points have been outlined in the Strengths and Weaknesses sections. If the authors can provide satisfactory responses and additional experiments addressing these points, I would consider revising my review.

---

> ### Author Response · Authors · 2023-11-18
> **Rebuttal + new experiments**
>
> Thank you for your thorough review. It seems the novelty of the contribution with respect to DPOK is your primary concern; as we explain in the top-level rebuttal, our paper was posted to arXiv 3 days before DPOK, and as such the work should be considered concurrent. The overall algorithm, narrative, and writing structure of our paper have not changed since that initial posting, and as such, our imitation of DPOK is impossible.
>
> That said, we have added an experiment directly comparing our method to DPOK in Appendix C of the updated PDF ([screenshot of quantitative results;](https://ibb.co/Jc0LmS0) [screenshot of qualitative results](https://ibb.co/z4shhBd)). Our method significantly outperforms the numbers reported by DPOK on the same task, even without KL regularization.
>
> > **out-of-domain evaluation is necessarily needed**
>
> Our comparison to DPOK includes the out-of-domain evaluation you recommended; namely, training using ImageReward and evaluating using Aesthetic Score. We find that both metrics increase across all prompts ([screenshot](https://ibb.co/Jc0LmS0)).
>
> > **Lack of code**
>
> Apologies for the oversight! Open-source code has been available for the past 6 months, but we forgot to provide an anonymized link in our submission. You can find anonymized code [here](https://anonymous.4open.science/r/ddpo-pytorch-1D5C/).
>
> > **the second noticeable difference is the automated generation of rewards using BLIP, which is already a standard engineering practice and has been claimed by RLAIF**
>
> We are not aware of any prior work that uses feedback from vision-language models to improve image-generative models. If you do know of such work, please let us know and we will add appropriate discussion!

---

> > ### Comment · Reviewer_qqX7 · 2023-11-18
> > **Thank you for the responses.**
> >
> > I thank the authors for the rebuttal, which has enhanced the article's validity and authenticity, and I am willing to update my rating.

---

> > > ### Author Response · Authors · 2023-11-18
> > >
> > > Hi reviewer qqX7,
> > >
> > > Thank you for the prompt response! Could you elaborate on what remaining concerns you have that bring the paper below the ICLR acceptance standards?

---

> ### Author Response · Authors · 2023-11-21
>
> Hi reviewer qqX7,
>
> To clarify, we believe that our rebuttal has thoroughly mitigated all of your stated weaknesses. Weaknesses (1) and (2) were about DPOK, which we have clarified is concurrent work, meaning that the DDPO algorithm should be considered an original intellectual contribution. Weakness (4) was about code, which we have provided. Weakness (3) was about out-of-domain reward evaluation, for which we have provided an additional experiment based on your recommendation.
>
> As such, if you still think this work is below the acceptance threshold for ICLR, we would greatly appreciate it if you could elaborate on why so that we can further improve our paper!
>
> Thank you,
> The Authors

---

> > ### Comment · Reviewer_qqX7 · 2023-11-21
> >
> > I am grateful to the author for the experimental responses during the Rebuttal period, which resolved some of my concerns. As a result, I have raised my score to 5. After a detailed reading and confirmation of the paper, I received two persistent inquiries from the author. Therefore, I will elaborate on my reasons for scoring as follows:
> >
> > Compared to concurrent work, this paper is so similar to DPOK in terms of innovation and paper writing but significantly weaker in contributions to reinforcement learning theory and insights than DPOK. Other reviewers, including rYFt, also raised questions about the similarity of this work. The paper was submitted after DPOK, and besides early convergence due to Important Sampling (I am not convinced by just one curve to claim this is a sufficient quantitative experiment, I am not satisfied with the experiment's sufficiency), I am not convinced that the paper has made sufficient innovative contributions to the community. Therefore, I have raised my score to 5, just below the acceptance threshold of ICLR.
> >
> > The authors repeatedly emphasize the simultaneity of the paper (not only in my thread) but fail to prove its superiority over DPOK through sufficient quantitative indicators. In the work of Lee et al., 'Optimizing DDPM Sampling with Shortcut Fine-Tuning', the method of using RL to fine-tune DDPM has been detailed, and MDP was fully introduced. This paper was published on Arxiv on 31 Jan 2023, well before its publication date. In my view, Lee et al. first introduced RL into DDPM in 'Optimizing DDPM Sampling with Shortcut Fine-Tuning,' and then in DPOK, systematically and rigorously proved the validity of the practices in 'Optimizing DDPM Sampling with Shortcut Fine-Tuning.' A similar work to this paper should be 'Optimizing DDPM Sampling with Shortcut Fine-Tuning,' not DPOK. Also, I believe the authors had sufficient time (about six months) to extend DPOK to better performance, yet claim it as concurrent work, and today, after six months, with insufficient experiments, prove its validity compared to DPOK. In my view, DPOK is a theoretical extension, while 'Optimizing DDPM Sampling with Shortcut Fine-Tuning' is **the similar work** of this paper.
> >
> > Regarding serious errors in the experiments. As the author mentioned in the appendix, there was a serious bug in the experiments of this paper, but they claimed that such a bug does not affect the fairness of the comparison. This bug was only responded to after community inquiries. That is, some results of the paper can only be achieved on customized devices like TPU, and I am skeptical about whether such bugs exist in other parts, so I reserve my judgment on the completeness of this paper.
> >
> > I do not think repeatedly asking and unilaterally resolving all my doubts is an effective Rebuttal method. Therefore, I will keep my rating.

---

> > > ### Author Response · Authors · 2023-11-21
> > >
> > > Thank you for the follow-up. While we do appreciate your perspective and would like to improve the paper to address any potential issues that you see, we disagree with a number of the points that you raised.
> > >
> > > Regarding shortcut fine-tuning (Fan et. al., 2023): as we wrote in our related work and methods sections, this work was the first to propose a policy gradient objective for training diffusion models. We believe we are quite up-front about this, but if this point is unclear, please let us know and we can revise it. That said, shortcut fine-tuning was aimed at a very different problem (improving fast sampling using a GAN-like objective) and only included small-scale experiments with unconditional diffusion models. In contrast, our work shows that RL can optimize a variety of different objectives, including prompt alignment with VLM-derived rewards. We believe that this is a significantly different contribution aimed at a different problem, and the experiments illustrate its flexibility and usefulness.
> > >
> > > Regarding DPOK: we recognize that there are similarities in innovation, writing, and experiments between our two papers. However, we emphasize that DPOK is concurrent because we want to make sure that these similarities do not affect the evaluation of our work. We did not think it was necessary to extend or improve DPOK, nor prove DDPO’s superiority over DPOK, because they are concurrent. Nevertheless, we found it valuable to include an additional experiment comparing DDPO to DPOK as a point of scientific interest, since our implementation achieves better performance than DPOK without DPOK's KL-regularization ([screenshot of results](https://ibb.co/Jc0LmS0)). This experiment was also a good addition since it evaluated a held-out reward, as you suggested, and showed that both the optimized reward and the held-out reward could improve significantly.
> > >
> > > Regarding the aesthetic predictor bug: as we have explained to reviewer rYFt, we have diligently re-run the primary comparison (Figure 4) after resolving the bug. The bug was caused by hardware-specific numerical instability, and caused the output of the aesthetic predictor to be perturbed in a way that affected the scale of the rewards and also the style of the resulting images (although they were still aesthetically pleasing, just in a different way). We value transparency and honesty in our research and note that the left-hand chart in the RWR interleaving ablation in the appendix still reflects the influence of the bug. However, the bug affected all methods in the same way, which is why we say that the quantitative comparison is fair. If the presence of this chart in our appendix causes any concerns, we are open to removing it, as it does not significantly impact the findings of our paper.
> > >
> > > **We would appreciate it if you could elaborate on the claim that “some results of the paper can only be achieved on customized devices like TPU.”** As far as we know, nothing in our submitted paper or our rebuttal suggests this. The anonymized code we provided ([link](https://anonymous.4open.science/r/ddpo-pytorch-1D5C/)) is written in PyTorch; we have successfully replicated our results on GPUs many times, and we welcome any further clarification on this matter.
> > >
> > > Finally, we would like to apologize if we came across as unilaterally resolving your stated weaknesses on your behalf. We only meant to summarize the ways in which we had addressed your concerns, hoping to receive elaboration on what you thought the remaining weaknesses were so that we could address them as well. In the end, our goal is improve the quality and scientific rigor of our paper as much as possible, and we are grateful for your input in this process.

---

### Author Response · Authors · 2023-11-18
**Summary of rebuttal + new experiments**

Hi all,

Thank you for your thorough and thoughtful reviews. We have updated the submission PDF with a few key changes, highlighted in blue. These include a number of new experiments to address the main concerns raised in the reviews.

**Comparison to DPOK**

As we mentioned in the related works section, DPOK is concurrent work. We say this because our paper was first posted to arXiv 3 days before DPOK.

That said, we have now run an experiment directly comparing to the DPOK tasks. You can find it in Appendix C of the updated submission ([screenshot of quantitative results;](https://ibb.co/Jc0LmS0) [screenshot of qualitative results](https://ibb.co/z4shhBd)). We found that DDPO outperforms the numbers reported in the DPOK paper, even without KL regularization. Since DPOK has not released code, we can’t determine what exactly the cause of this is, but it does suggest that the particular design decisions in our implementation of DDPO are more effective for RL training of diffusion models.

**Code Release**

Open-source code has been available for the past 6 months, but we forgot to provide an anonymized link in our submission. Apologies for the oversight; you can find anonymized code [here](https://anonymous.4open.science/r/ddpo-pytorch-1D5C/).

**Classifier Guidance Baseline**

For a differentiable reward function such as the LAION aesthetic predictor, we agree that a classifier guidance style baseline makes sense. We ran an experiment applying [universal guidance](https://arxiv.org/abs/2302.07121) to the aesthetic predictor, which you can find in Appendix B of the updated submission ([screenshot](https://ibb.co/dfJbVTw)). We found that universal guidance is able to produce a non-negligible increase in aesthetic score, but it is much smaller than DDPO finetuning while also being prohibitively slow (2 minutes per image on an A100) due to the backpropagation of gradients through the large guidance network.

---

### Meta-Review · Area_Chair_ZNTV · 2023-12-05

**Metareview:**

The paper proposes an RL-based method for fine-tuning large-scale diffusion models to match potentially nondifferentiable reward functions.   the theoretical formulation and empirical evaluation are sufficient according to the reviewers and my judgment.

Most of the reviewers initially thought the paper was similar to DPOK and worried about the novelty. As claimed by the authors, this paper is a concurrent work as DPOK and was presented on Arxiv earlier. Given this information as well as the rebuttal, most of the reviewers changed their minds and raised the scores.

However, Reviewer qqX7 still maintains reservations regarding the A+B contribution style and harbor concerns about the potential overoptimization risk. Yet, qqX7 also said, "*I would not be opposed to accepting the paper*".

Considering all the information, I tend to accept this paper as a poster.

**Justification For Why Not Higher Score:**

Reviewer qqX7 still maintains reservations regarding the A+B contribution style and harbor concerns about the potential overoptimization risk.

**Justification For Why Not Lower Score:**

The theoretical formulation and empirical evaluation are sufficient according to the reviewers and my judgment.

---

### Decision · Program_Chairs · 2024-01-16

Accept (poster)